# Pattern of Radiotherapy Treatment in Low-Risk, Intermediate-Risk, and High-Risk Prostate Cancer Patients: Analysis of National Cancer Database

**DOI:** 10.3390/cancers14225503

**Published:** 2022-11-09

**Authors:** Rishabh Agrawal, Asoke Dey, Sujay Datta, Ana Nassar, William Grubb, Bryan Traughber, Tithi Biswas, Roger Ove, Tarun Podder

**Affiliations:** 1Department of Radiation Oncology, Medical College of Georgia, Augusta, GA 30912, USA; 2Department of Management, University of Akron, Akron, OH 44325, USA; 3Department of Statistics, University of Akron, Akron, OH 44325, USA; 4Department of Radiation Oncology, Case Western Reserve University School of Medicine, Cleveland, OH 44106, USA; 5Mayo Clinic, Rochester, MN 55905, USA; 6University Hospitals Seidman Cancer Center, Cleveland, OH 44106, USA

**Keywords:** prostate cancer, overall survival, radiotherapy, brachytherapy, IMRT, proton-beam therapy, SBRT

## Abstract

**Simple Summary:**

Prostate cancer (PCa) is the most common cancer and the second leading cause of cancer-related mortality among males in the US. Definitive radiation therapy (RT) plays an important role in curative-intent treatment for localized PCa and can be delivered with several different techniques, depending on the availability of resources and patient-specific criteria. With an analysis of the extensive National Cancer Database, this paper investigates trends in utilization, survival probability, and factors associated with overall survival of six common RT modalities utilized for the treatment of PCa patients—stratified by the three risk groups.

**Abstract:**

Background: In this study, the utilization rates and survival outcomes of different radiotherapy techniques are compared in prostate cancer (PCa) patients stratified by risk group. Methods: We analyzed an extensive data set of N0, M0, non-surgical PCa patients diagnosed between 2004 and 2015 from the National Cancer Database (NCDB). Patients were grouped into six categories based on RT modality: an intensity-modulated radiation therapy (IMRT) group with brachytherapy (BT) boost, IMRT with/without IMRT boost, proton therapy, stereotactic body radiation therapy (SBRT), low-dose-rate brachytherapy (BT LDR), and high-dose-rate brachytherapy (BT HDR). Patients were also stratified by the National Comprehensive Cancer Network (NCCN) guidelines: low-risk (clinical stage T1–T2a, Gleason Score (GS) ≤ 6, and Prostate-Specific Antigen (PSA) < 10), intermediate-risk (clinical stage T2b or T2c, GS of 7, or PSA of 10–20), and high-risk (clinical stage T3–T4, or GS of 8–10, or PSA > 20). Overall survival (OS) probability was determined using a Kaplan–Meier estimator. Univariate and multivariate analyses were performed by risk group for the six treatment modalities. Results: The most utilized treatment modality for all PCa patients was IMRT (53.1%). Over the years, a steady increase in SBRT utilization was observed, whereas BT HDR usage declined. IMRT-treated patient groups exhibited relatively lower survival probability in all risk categories. A slightly better survival probability was observed for the proton therapy group. Hormonal therapy was used for a large number of patients in all risk groups. Conclusion: This study revealed that IMRT was the most common treatment modality for PCa patients. Brachytherapy, SBRT, and IMRT+BT exhibited similar survival rates, whereas proton showed slightly better overall survival across the three risk groups. However, analysis of the demographics indicates that these differences are at least in part due to selection bias.

## 1. Introduction

Prostate cancer (PCa) is one of the most common cancers among men, accounting for slightly above 21% of new cancer diagnoses in men yearly (~250,000 new cases yearly) [1]. Despite a high incidence rate, prostate cancer only accounts for 10% of direct primary cancer-related deaths in men and boasts roughly a 98% survival rate [2]. Mortality in PCa patients is often secondary to issues unrelated to the original cancer and rather due to negative effects of treatment modalities, with consequences such as respiratory failure and cardiac disease [3]. PCa patients are commonly stratified into three risk groups (low, intermediate, and high) based on three main factors: clinical T-stage, Gleason score (now histologically compressed into Grade Groups), and prostate-specific antigen (PSA) level [4].

A wide array of treatment modalities is currently available, and multiple considerations play into the treatment decision, such as age, risk group, demographic, treatment center, and personal preferences, among other factors. Surgery (prostatectomy), brachytherapy (BT), intensity-modulated radiation therapy (IMRT), hormonal therapy, or a combination of multiple modalities are some of the treatment options commonly offered [5]. Clinical outcomes of various treatment modalities have proven each of them to be safe, giving patients increased autonomy to choose a treatment plan of their preference [6]. However, this may be limited by the availability of treatment options at a center, as well as physician preference [6,7,8]. Multidisciplinary teams of radiation oncologists, clinical oncologists, and urologists often play a role in determining treatment options, considering the patients’ comorbidities and pathologic characteristics of tumors [9].

IMRT is widely used as the standard radiotherapy for the management of prostate cancer [10]. Alongside proton beam therapy (PBT), IMRT’s normal tissue sparing permits dose escalation that is advantageous in cancer-control rates without increases in toxicity [11,12,13]. There is evidence that high-dose IMRT (up to 81 Gy) displays high efficacy in preventing biochemical failure, with acceptable side effects over the course of 10 years [14]. Similarly, the normal tissue-sparing characteristic of IMRT delivers a high dose per fraction with tolerable risk, allowing for hypo-fractionated treatment of PCa patients, a technique that holds promise for better outcomes [15].

Though initially utilized as a boost treatment modality to external beam radiation therapy (EBRT), high-dose-rate BT (BT HDR) as a monotherapy proved to be successful in overcoming some limitations of EBRT [16,17,18]. BT HDR is advantageous in overcoming overall organ motion and sparing nearby organs with a rapid dose fall-off. Coupled with its high dose conformity within the target volume, BT HDR allows for thorough and concise biological planning with minimal dosimetric uncertainty [16]. There is also a relatively short treatment period with excellent functional outcomes [10]. Even though BT is important in treating localized, low-risk prostate cancer, it is utilized neither frequently nor uniformly across nationwide practices [16,19,20]. However, its success as a monotherapy in patients with PCa has sparked interest in and clinical justification for the use of other hypo-fractionated radiation therapies, such as stereotactic body radiation therapy (SBRT) [17,21].

Compared to IMRT, stereotactic body radiation therapy (SBRT) is an attractive alternative that administers a higher dose per fraction with a fewer number of fractions [22]. This raises the concern for toxicity—particularly genitourinary (GU), gastrointestinal (GI), and sexual dysfunction in the case of prostate cancer [9,22]. However, early reports from various prospective studies have indicated that GI/GU toxicities in SBRT are comparable to other modalities of RT [23,24].

Though PBT is still relatively new, its intrinsic advantage in sparing normal tissues and organs at risk (bladder, rectum, etc.) has increased its popularity as a treatment modality [25,26]. Some authors have reported that PBT has been associated with higher overall survival compared to EBRT and with similar overall survival outcomes to BT [27]. Even though many of the dosimetry studies have shown overall lower radiation to normal tissues and theoretically higher effectiveness for PBT, no study could show a clear benefit over traditional photon-based treatments such as IMRT [7,27,28].

There have been a few randomized clinical trials to investigate the effectiveness of treatments for prostate cancer. Much of the available clinical data remains inconclusive due to the lack of concrete control groups and the possibility of confounding variables [29,30]. There were also some attempts to assess the effectiveness of the different treatment modalities, but such studies were prematurely closed due to decreases in longitudinal active surveillance and thus potential resulting toxicity-related concerns [31]. Hence, randomized data directly comparing the efficacies and outcomes of survival for various RT techniques are currently lacking on a national scale [20]. 

In this paper, we retrospectively investigate the trends of the usage of six major RT modalities, their associations with certain variables, and overall survival outcomes. It is our aim to detail a representative view of prostate cancer management in the US population (i.e., a real-world scenario).

Using the NCDB, we accumulated data from an extensive list of centers across the country that vary in factors such as demographics served, size of the institution, research intent, etc. This large database may eliminate some of the biases expressed by big cancer centers and instead help uncover the outcomes in small, community-based cancer centers.

## 2. Materials and Methods

### 2.1. Data Source

The NCDB is a nationwide clinical oncology database cosponsored by the American College of Surgeons and the American Cancer Society. Data are collected from the hospital registries of 1500+ cancer-accredited facilities and represent an estimated 70% of all cancer cases across the United States. The information pulled from the NCDB in this project includes information such as patient demographics, facility type or location, cancer characteristics, Charlson–Deyo scores, treatment modality, and survival data for prostate cancer patients from 2004 to 2015. The records of these patients in the database are de-identified and sent to researchers for analysis after acceptance for related projects. The American College of Surgeons and the Commission on Cancer have not verified and are not responsible for the conclusions drawn from the data by the investigators in this study.

### 2.2. Subject Selection

There were initially 1,380,357 patients identified in the NCDB database that were diagnosed with prostate cancer from 2004 to 2015. Of this total, only those patients with PSA levels between 0.2–97.9 ng/mL, a Gleason score between 2–10, and a clinical stage defined as 1, 2, 3, 4, 2A, or 2B were considered, yielding 985,197 subjects. Only patients with AJCC N0 and M0 were considered for the study; the rest were excluded (Figure 1). From the remaining 877,700 subjects, patients were selected for this study who had one of the six modalities with a reasonable radiation dose: (1) 40–55 Gy IMRT Initial + BT boost, (2) 65–85 Gy IMRT Initial (with consideration of 20–40 Gy IMRT boost), (3) 65–85 Gy Proton, (4) 30–50 Gy SBRT, (5) BT LDR, and (6) BT HDR. This reduced the total sample size to 216,714 subjects. No radiation boost was considered for 65–85 Gy IMRT, proton, SBRT, BT LDR, and BT HDR. Monotherapy was considered for LDR, HDR, or unspecified brachytherapy. Following the selection of treatment modalities, PCa patients that had undergone chemotherapy, prostatectomy surgery, or with unknown chemotherapy/surgery status were also excluded; finally, a total of 199,926 patients were eligible for this study (Figure 1). Total subject frequency for each of the treatment groups can be found by low-, intermediate-, and high-risk groups in Figure 2, Figure 3 and Figure 4 and Table 1 and Table 2.

Total frequencies of these patients were stratified by risk according to the NCCN guidelines into the three categories to be studied: low-risk (clinical stage T1–T2a, Gleason Score (GS) ≤ 6 (Grade Group 1), and PSA < 10 ng/mL); intermediate-risk (clinical stage T2b–T2c, or GS = 7 (Grade Groups 2 and 3), or PSA = 10–20 ng/mL); and high-risk (clinical stage T3–T4, or GS = 8–10 (Grade Groups 4 and 5), or PSA > 20 ng/mL). To be eligible for these risk categories, patients had to meet all three necessary criteria. This is with the exception that other T-staged patients (i.e., T2-undefined) that met the GS and PSA criteria were considered in their respective risk groups.

### 2.3. Definition of Variables

The years of diagnosis were grouped into periods from 2004–2007, 2008–2011, and 2012–2015. Age was stratified into groups of under 65, 65–69, 70–74, and over 74 years old. The patient’s race was defined as white, black, other, or unknown. Insurance was categorized as private, government (including Medicare, Medicaid, and other government), or no insurance. The Charlson–Deyo comorbidity index was recorded as the summation of comorbid conditions and was scored as 0, 1, 2, with a score of 0 representing no comorbid conditions recorded [32]. The 2013 U.S. Department of Agriculture Rural-Urban Continuum was used to define metropolitan, urban, and rural areas. Counties in metropolitan areas were coded as metropolitan, counties with an urban population of ≥2500 but not in a metropolitan area were termed urban, and counties with an urban population of <2500 were termed rural. Residential areas were stratified by median income into less than $38,000, $38,000–$47,999, $48,000–$62,999, and $63,000 and above; a small set of patients’ statuses was unknown. Similarly, the education level of the residential areas was clustered by the percentage of residents without a high school degree: <7%, 7–12.9%, 13–20.9%, >21%, and unknown. Distance from residence to the facility was calculated using the center of the patient’s zip code to the treating facility’s mailing address. Facilities were primarily separated on whether they were classified as academic/research-based or non-academic, with the status of 10 facilities unknown. Facility location was defined as Northeast: CT, MA, ME, NH, NJ, NY, PA, RI, and VT; South: AL, AR, DC, DE, FL, GA, KY, LA, MD, MS, NC, OK, SC, TN, TX, VA, and WV; Midwest: IA, IL, IN, KS, MI, MN, MO, ND, NE, OH, SD, and WI; and West: AZ, AK, CA, CO, ID, HI, MT, NM, NV, OR, UT, WA, and WY. Patients were classified based on their T-stage into Stage 1, 2, 2A, 2B, 3, 4, or unknown. Varying PSA levels (from 0.2 to greater than 74.9 ng/mL) were used to stratify patients into separate categories. Similarly, there was the utilization of the Gleason Score to divide patients into categories of <6, 6, 7, 8, 9, and 10. The use of hormone therapy towards PCa was of interest in the form of yes, no, or unknown.

### 2.4. Statistical Analysis

The main aims of this study were the trend of the utilization of radiation treatment modalities and the median overall survival (OS). Baseline characteristics, defined as per the variables above, allowed for univariate and multivariate Cox proportional hazard models for each risk category. Relevant covariates included in the Cox model are years of diagnosis, age, race, insurance status, Charleson–Deyo morbidity index, residential setting, median income, distance from facility to residence, facility type, facility location, hormonal therapy, and radiotherapy modality (the key independent variable). PSA is not considered in the Cox model, as PSA scores are already included in the definition of risk groups. The respective unadjusted and adjusted hazard ratios (HR) then allowed for a direct comparison of survival outcomes among different categories. Outcome survival probabilities were determined using Kaplan–Meier estimator. IBM SPSS software (Version 26.0. Armonk, NY, USA: IBM Corp) was used for overall statistical analysis. For this study, a *p*-value < 0.05 was considered statistically significant.

### 2.5. Disclaimer

This study was carried out in accordance with the guidelines, data dictionary, and accompanying de-identified files provided by the NCDB. The findings of this study are not backed by the NCDB and are independently concluded by the listed investigators.

## 3. Results

### 3.1. Gross Breakdown

A total of 199,926 patients were studied after all pertaining inclusion and exclusion criteria as described above. The distribution of the total sample is presented in Table 1. The breakdown of this large NCDB sample was first via risk stratification: 71,146 low-risk subjects, 84,741 intermediate-risk subjects, and 44,039 high-risk subjects. A further breakdown of these risk categories into the various treatment modalities is provided at the end of Table 2 and with greater detail in Table 3, Table 4 and Table 5. Based on the specific radiotherapy techniques used, we considered multiple treatment groups of patients in the eligible sample (*n* = 199,926; 100%) inclusive of all three aforementioned risk groups: (1) IMRT Initial + BT boost (*n* = 12,734; 6.4%), (2) IMRT Initial with/without IMRT boost (*n* = 106,246; 53.1%), (3) proton (*n* = 4561; 2.3%), (4 SBRT (*n* = 7533; 3.8%), (5) BT LDR (*n* = 45,452; 22.7%), and (6) BT HDR (*n* = 23,400; 11.7%). Year-to-year trends of the total sample can also be visualized in Figure 2, Figure 3 and Figure 4, separated by risk category.

Baseline characteristics, such as specific case features, patient types, and facility descriptors, among others, have also been presented by risk categories in Table 2. These baseline characteristics for the three risk groups are broken down more comprehensively by each treatment modality in Table 3, Table 4 and Table 5.

### 3.2. Survival Outcomes

Kaplan–Meier survival curves for the low-, intermediate-, and high-risk PCa patients are presented in Figure 5, Figure 6 and Figure 7, respectively. It is observed that patterns of treatment effectiveness across the three risk groups are largely similar: proton exhibited a slightly better outcome. Meanwhile, the survival probabilities of the other four modalities (IMRT+BT, SBRT, BT LDR, and BT HDR) are very similar to one another. It is noted that although IMRT was the main treatment modality for high-risk patients (74.6%), compared to the rest of the patients being treated with the other five modalities combined (25.4%), the outcome pattern was mostly similar except for a slightly diminishing difference between proton and other modalities. A closer look into these plots reveals that 100 months OS probability is in the range of 80–92% for low-risk patients (Figure 5). The 100 months OS is estimated between 65–88% and 61–78% for intermediate-risk (Figure 6) and high-risk patients (Figure 7), respectively.

### 3.3. Univariable Analyses of Patient Population

Univariable analysis was conducted by stratifying patients into low-, intermediate-, and high-risk groups, then considering a multitude of factors regarding demographics, disease characterization, and treatment methods. Statistically significant positive and negative associations that were noted between the studied characteristics and overall survival are listed below (Table 6, Table 7 and Table 8).

In low-risk PCa populations, overall survival was significantly associated with patient races other than black or white and with increasing median incomes greater than $38,000. Patients living 10–25+ miles away from a treatment facility also demonstrated increased survival, whereas those living in residential areas, with an increasing percentage of the population without high school diplomas, showed inferior outcomes. Furthermore, locations and types of treatment facilities were found to be significant factors: overall survival was significantly associated with treatment at academic or research facilities (vs. non-academic), along with those facilities located in the West. Regarding disease and subsequent treatment, the following factors held negative associations with overall survival: patients with increasing scores on the Charlson–Deyo comorbidity index and those that had hormonal therapy. The overall survival of proton therapy was closely followed by IMRT + BT boost, BT HDR, SBRT, and BT LDR. Lastly, increasing patient ages (greater than 65) and rural/urban residential settings (vs. metro) were associated with poorer outcomes.

In intermediate-risk PCa populations, overall survival was significantly associated with patient races other than black or white and with increasing median incomes greater than $48,000. Those living 5–25+ miles from treatment facilities, academic or research facility type, and facilities located in the West were also associated with better outcomes. Similar to low-risk PCa populations, overall survival in proton therapy was closely followed by IMRT + BT boost, BT HDR, BT LDR, and SBRT. Increasing patient ages (greater than 65), urban/rural residential settings (vs. metro), residential areas with an increasing percentage of the population without a high school degree (greater than 7%), government-insured, increasing Charlson–Deyo comorbidity index (greater than 0), and the use of hormone therapy were associated with inferior outcomes.

In high-risk PCa populations, patients identifying as black or other, those with median incomes greater than $63,000, those living 25+ miles from a facility, and those treated at academic or research facilities were significantly associated with better overall survival. Like low-risk and intermediate-risk PCa patients, in high-risk PCa populations, the outcome of proton therapy was followed by IMRT + BT boost, BT HDR, and BT LDR. Increasing age (greater than 65) and residential areas with an increasing percentage of the population who do not have a high school degree were associated with poorer outcomes. With regards to disease and subsequent treatment, an increasing Charlson–Deyo comorbidity index (greater than 0), T-stages 2A, 3, and 4, and the use of hormonal therapy were all associated with inferior outcomes.

### 3.4. Multivariable Analyses of Patient Populations

Multivariable analysis of the patient population was stratified into low-, intermediate-, and high-risk PCa patients. Statistically highly significant positive and negative associations with overall survival are discussed below (Table 6, Table 7 and Table 8).

In low-risk PCa populations, median incomes greater than $63,000, those living 25+ miles from the facility, and facilities located in the West were positively associated with overall survival. Increasing age over 65, increasing Charlson–Deyo comorbidity index greater than 0, and residential areas with an increasingly greater percentage of the population without high school degrees (greater than 7%) were associated with poorer overall survival. In the multivariate analysis of the low-risk PCa population, the overall survival of proton therapy was closely followed by IMRT + BT boost, BT HDR, BT LDR, and SBRT (Table 6).

In intermediate-risk PCa populations, patient races other than black or white, median incomes of greater than $63,000, and facilities located in the West were associated with better outcomes. Increasing age (over 65 years old), increasing Charlson–Deyo comorbidity index (greater than 0), and residential areas with increasing percentages of the population without high school degrees (greater than 13%) were associated with inferior survival. Similar to the multivariable analysis of the low-risk PCa population, overall survival of proton therapy in intermediate-risk PCa patients was closely followed by IMRT + BT boost, BT HDR, and BT LDR (Table 7).

In high-risk PCa populations, those living 25+ miles from a facility were associated with better outcomes. Increasing age (greater than 65), increasing Charlson–Deyo comorbidity index (greater than 0), and T-stages greater than three were associated with inferior overall survival. Contrary to the other two risk groups regarding radiotherapy, IMRT + BT boost was most associated with overall survival, followed by BT HDR and BT LDR (Table 8).

## 4. Discussion

To the best of our knowledge, this study was the first to (1) break down the usage of these six common radiation therapy modalities (IMRT, IMRT + BT boost, proton, SBRT, BT LDR, and BT HDR) in the treatment of prostate cancer at varying risks on a national scale, (2) assess different variables and their corresponding associations with survival by utilizing hazard ratio models, and (3) investigate survival outcome and pattern of utilization concerning each treatment modality and risk category. We hope the results from the analyses pertaining these three aims have provided some useful insights for radiotherapy-based prostate cancer management in the general US population.

The pattern of utilization in these six radiation treatment modalities, across risk categories, was the first notable set of results identified in this study (Table 1). In the studied PCa patient population from the 2004–2015 NCDB database, IMRT was the most common modality in intermediate-risk (48,160; 56.8%), high-risk (32,851; 74.6%), and overall (106,246; 53.1%) patient groups. We speculate that IMRT’s popularity is due to its being a commonly available treatment modality and its toxicity control by sparing critical structures [7,15,33].

Additionally, proton therapy was discovered to be the least commonly used modality within all risk groups, with the following breakdown: low-risk (1777; 2.5%), intermediate-risk (2296; 2.7%), high-risk (488; 1.1%), and overall (4561; 2.3%). We speculate that low usage of PBT could be attributed to its lack of availability and possibly due to its higher cost of treatment [7]. With this in mind, the slightly better overall survival in proton therapy patients could be associated with higher socioeconomic status, higher education, and access to better healthcare; these factors need to be investigated in future studies. 

Usage of the modalities was further broken down by year of diagnosis to provide insight into any dynamic trends over the study period in Figure 2, Figure 3 and Figure 4. Such trends include a relative increase in yearly utilization of IMRT over the study period in all three risk groups; meanwhile, BT LDR was noted to have decreasing usage in the treatment of all three risk groups over the same period. There was a steady increase in SBRT utilization, especially for low-risk and intermediate-risk patients, whereas the utilization of BT HDR declined over time.

Next, this study allowed us to analyze the rates and associations with survival outcomes of some variables in our studied population. Increasing median incomes were associated with better overall survival in all three disease/risk stratifications in both the univariable and multivariable analyses (Table 6, Table 7 and Table 8). Academic/research facilities were also associated with increased overall survival. As expected, increased Charlson–Deyo comorbidity index scores and T-stage classification were associated with inferior outcomes. To our surprise, increased distance between residence and treatment facilities confirmed better outcomes, to some degree, for almost all risk groups, except for the intermediate-risk population in the multivariable analysis. This finding contrasts with a previous registry-based study of breast cancer patients that found those living 40+ miles from an RT facility were significantly less likely to receive the entire radiotherapy course as planned, thus negatively affecting outcomes [34].

In each disease risk group, under univariable and multivariable analyses, overall survival of proton therapy was closely followed by IMRT+ BT boost. However, in the multivariable analysis of high-risk PCa populations, IMRT + BT boost was most associated with highest overall survival. Interestingly, the high rates of government insurance in the patient sample may have also played a part in the usage of certain treatment modalities. In Medicare beneficiaries with a first-time diagnosis of prostate cancer from 2006 to 2016, utilization of IMRT increased the most (23.6%, *p* < 0.0001), whereas PBT increased the least (0.74%, *p* < 0.0001) [35].

Though it is possible that some of the studied variables may have direct causation with survivability, this type of conclusion cannot be justified by our study design and the available data. Instead, it is plausible that there are further factors that link the considered variables and hazard ratios in this study. One such example would be general differences in radiotherapy modalities offered by region of the country or by facility type. Our study finds differences in the offering of radiotherapy modalities, such as proton therapy being more often utilized in the West than other regions of the country and IMRT and BT modalities being offered more often in non-academic centers. Findings from similar registry-based studies have hypothesized that areas with a higher density of radiation oncologists may have more competitive market influences that drive demand for greater technological availability and variation of RT modalities, and thus may misrepresent true survival outcomes [36,37]. Hence, the variables investigated in the univariable and multivariable analyses are significant but cannot be definitively linked as individualized causes to the hazard ratios.

Though not particularly investigated in this study, we believe that the application of radiomics, i.e., the extraction and analysis of quantitative imaging findings from radiographic images, would provide additional critical factors that need to be studied in the future. We hypothesize that variations in imaging modalities and quantitative analysis in imaging may similarly hold significant associations with overall survival [38,39].

As observed in Figure 5, Figure 6 and Figure 7, though the survival probability was slightly better in proton therapy for low- and intermediate-risk groups, IMRT + BT boost performed fairly well across the three risk groups. The patient selection bias and other compounding factors can be ruled out in the case of the proton therapy. The other four modalities performed similarly to one another, midway between the results for proton radiotherapy and IMRT only. It is interesting to note that a large number of low-risk patients had hormonal therapy, which is not commonly recommended. This use was not strongly correlated with brachytherapy patients, for whom androgen ablation is commonly used for prostate volume reduction.

Due to inherent biases in analyzing retrospective data, the results from this study should not be used to replace randomized clinical trials that test treatment effectiveness [10]. At the same time, we would like to acknowledge some of the other limitations that also may have been minimized, yet are inherently present in this study: (1) There may be an inherently misrepresented risk stratification of PCa patients built into the NCDB, due to differences in PSA level acquisition and calibration methods resulting from poor harmonization of laboratory methods [40]. (2) There may be a lack of uniform information/guidelines surrounding treatment choices in PCa patients. Previous residency training experience, interaction/advocation by third-party vendors [41], and general hesitance for BT by urologists and patients [10]—among other factors that can lead to treatment selection bias built into the dataset. (3) Specific follow-up patterns and timeframes are not reported, possibly influencing survival outcomes in cases of adjuvant therapy or recurrence. (4) There is an inherent selection bias, as only accredited hospitals input data into the NCDB. (5) Though the study captures an objective survival standard, there may have been differences in other important clinical endpoints (quality of life, adverse effects, etc.). (6) There are no toxicity-related data from the NCDB. (7) Though this study only analyzes non-surgery subjects, real-life clinical management of PCa patients is often inclusive of those techniques and thus may not be a complete representation.

Though we acknowledge the potential limitations in this study, there are plenty of strengths that we would also like to highlight: (1) By utilizing the NCDB entries, our data set is representative of real-world practices from a large number of institutions across the country, where the majority of PCa patients are treated. Having a nationwide patient cohort eliminates many of the biases and confounding variables that may be present in single-institution studies. (2) The total sample was stratified into three risk categories: low, intermediate, and high. This allows clinicians to have more evidence in choosing treatments that may be tailored to future patients’ risk categories. (3) There were objective and reproducible criteria placed for the separation of the patient sample into the three risk groups. (4) Survival is defined as the most important outcome of treatment according to 2017 NCCN guidelines and thus was used as the primary endpoint in this study [42]. Despite these strengths, the presence of limitations suggests further investigation is warranted.

## 5. Conclusions

This large-scale analysis of the NCDB revealed that multiple differing treatment patterns existed for non-surgical prostate cancer patients from 2004 to 2015. Stratification via risk-group categories showed some modalities to be preferentially used in the management of PCa patients. The rate of utilization for each radiation therapy modality was not representative of its corresponding median OS in the studied population. This discrepancy may have resulted from selection bias and confounding factors, both by the clinicians’ offering of select modalities and the patients’ abilities to pursue a certain therapy. Several social factors were noted to be associated with higher overall survival: higher median income, receipt of treatment at academic/research facilities, and increased distance between patient’s residence and treatment facility. Future studies of PCa radiotherapy modalities should be conducted with randomized clinical trials with a longer patient follow-up.

## Figures and Tables

**Figure 1 cancers-14-05503-f001:**
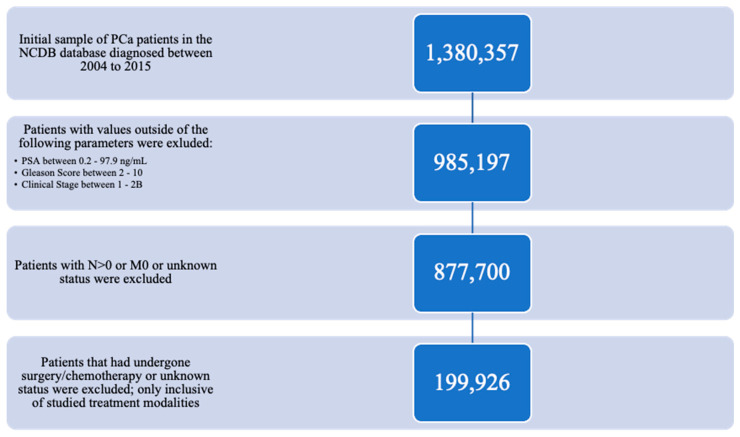
CONSORT diagram—selection of the prostate cancer patient population from NCDB 2004–2015.

**Figure 2 cancers-14-05503-f002:**
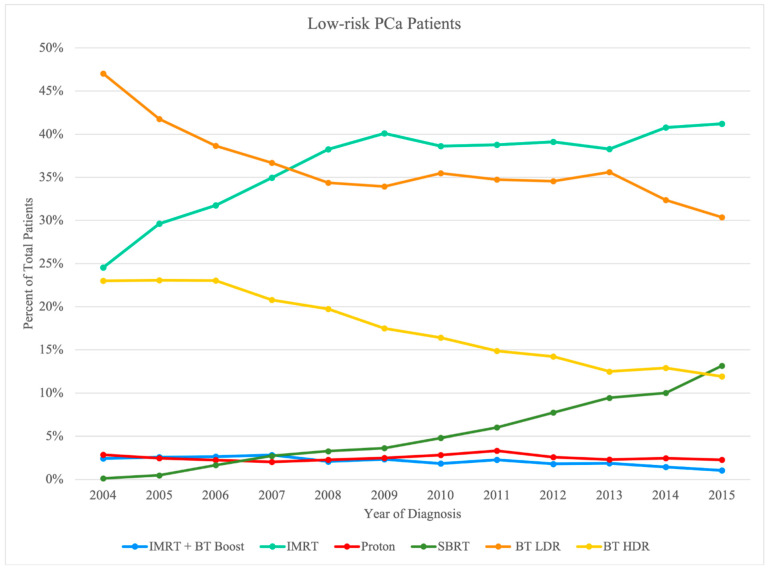
Pattern of utilization of radiation therapy modality for low-risk PCa by treatment modality.

**Figure 3 cancers-14-05503-f003:**
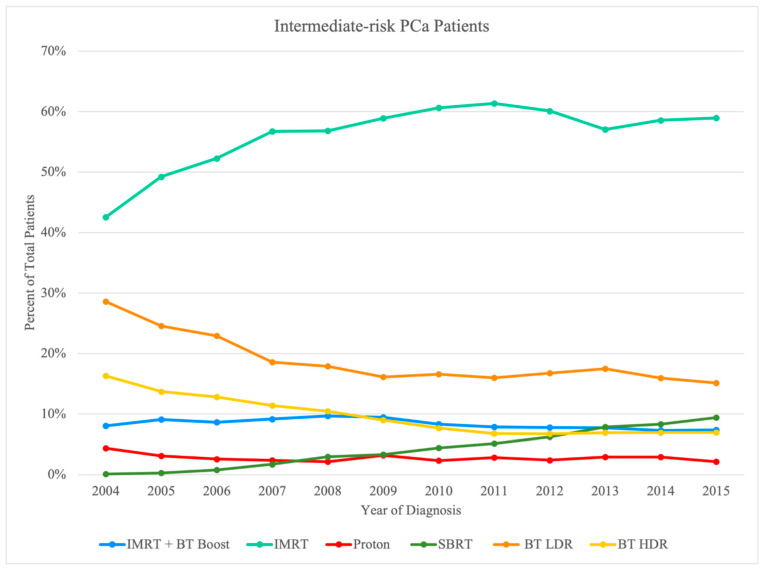
Pattern of utilization of radiation therapy modality for intermediate-risk PCa patients by treatment modality.

**Figure 4 cancers-14-05503-f004:**
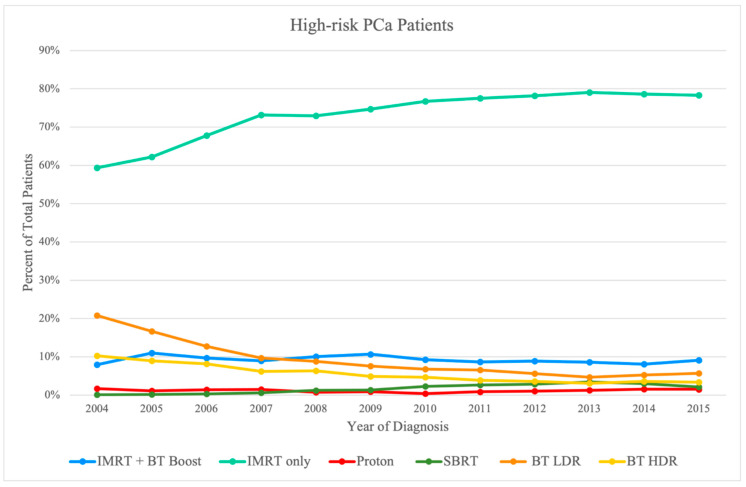
Pattern of utilization of radiation therapy modality for high-risk PCa patients by treatment modality.

**Figure 5 cancers-14-05503-f005:**
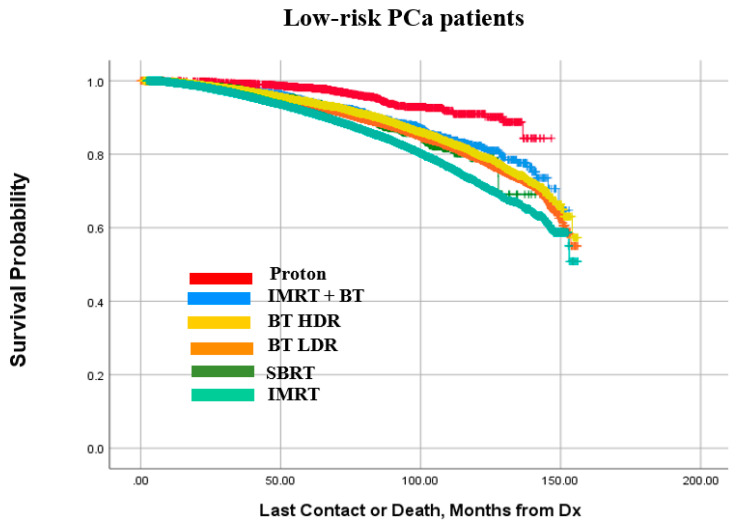
Kaplan–Meier curve for cumulative survival of low-risk patient prostate cancer (PCa) patients treated with the compared modalities.

**Figure 6 cancers-14-05503-f006:**
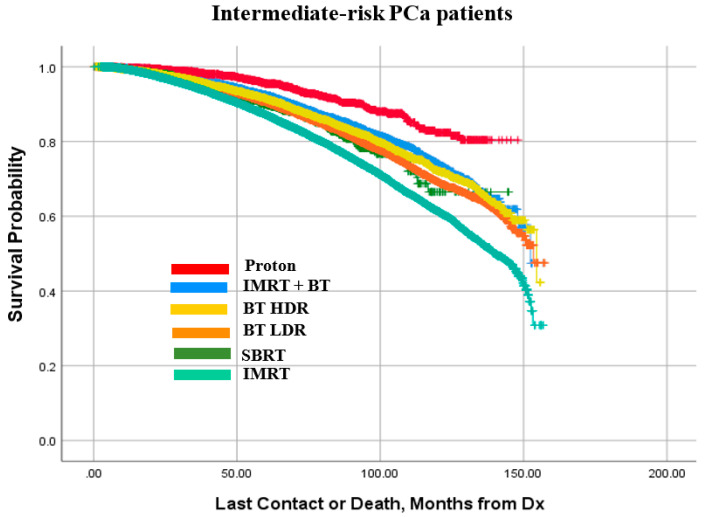
Kaplan–Meier curve for cumulative survival of intermediate-risk patient prostate cancer (PCa) patients treated with the compared modalities.

**Figure 7 cancers-14-05503-f007:**
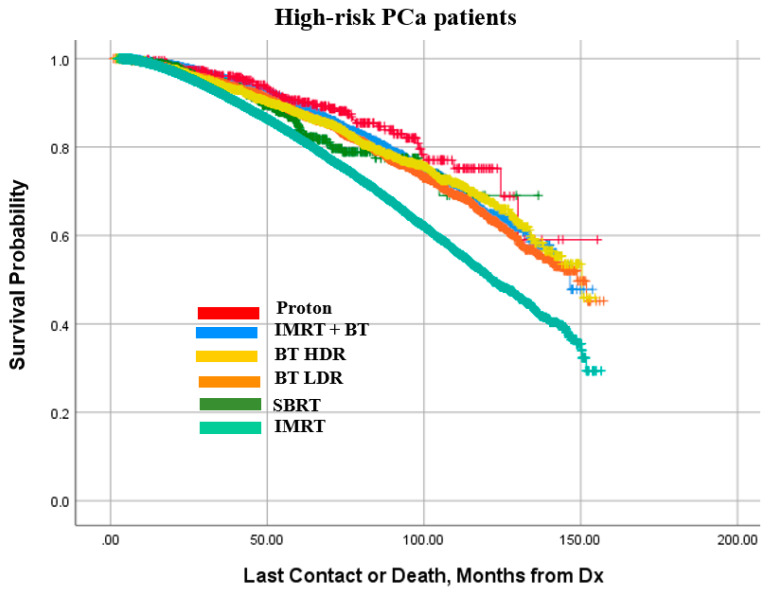
Kaplan–Meier curve for cumulative survival of high-risk patient prostate cancer (PCa) patients treated with the compared modalities.

**Table 1 cancers-14-05503-t001:** Stratification of studied PCa population in NCDB Database (2004–2015) by research risk groups and treatment modality.

Modality	Research Risk Groups			Total
Patient Selection	Low Risk	Intermediate Risk	High Risk	
IMRT + BT Boost	1604	7092	4038	12,734
IMRT only	25,235	48,160	32,851	106,246
Proton	1777	2296	488	4561
SBRT	2918	3795	820	7533
BT LDR	26,311	15,534	3607	45,452
BT HDR	13,301	7864	2235	23,400
Total	71,146	84,741	44,039	199,926

**Table 2 cancers-14-05503-t002:** Baseline characteristics of all patients with low-risk, intermediate-risk, and high-risk prostate cancer who received the research treatments.

Baseline Characteristics	Low-Risk Patients (%)	Intermediate-Risk Patients (%)	High-Risk Patients (%)
Overall	71,146	84,741	44,039
Year of Diagnosis			
2004–2007	30,987 (43.6)	24,365 (28.8)	10,821 (24.6)
2008–2011	26,427 (37.1)	31,019 (36.6)	16,325 (37.1)
2012–2015	13,732 (19.3)	29,357 (34.6)	16,893 (32.6)
Age (years)			
<65	29,801 (41.9)	24,696 (29.1)	10,603 (24.1)
65–69	17,959 (25.2)	20,160 (23.8)	9216 (20.9)
70–74	14,760 (20.7)	20,776 (24.5)	10,606 (24.1)
>74	8626 (12.1)	19,109 (22.5)	13,614 (30.9)
Race			
White	59,154 (83.1)	68,724 (81.1)	35,131 (79.8)
Black	9665 (13.6)	12,920 (15.2)	7201 (16.4)
Other	1533 (2.2)	2217 (2.6)	1292 (2.9)
Unknown	794 (1.1)	880 (1.0)	415 (0.9)
Insurance Status			
None	30,614 (43.0)	27,140 (32.0)	11,823 (26.8)
Government	38,886 (54.7)	55,566 (65.6)	31,059 (70.5)
Private	700 (1.0)	953 (1.1)	617 (1.4)
Unknown	946 (1.3)	1082 (1.3)	540 (1.2)
Charlson-Deyo Comorbidity index			
0	62,365 (87.7)	72,396 (85.4)	37,346 (84.8)
1	7625 (10.7)	10,157 (12.0)	5413 (12.2)
2+	1156 (1.6)	2188 (2.6)	1280 (2.9)
Residential Setting			
Metro	57,607 (81.0)	68,094 (80.4)	35,051 (79.6)
Urban	10,393 (14.6)	12,617 (14.9)	6833 (15.5)
Rural	1350 (1.9)	1851 (2.2)	982 (2.2)
Unknown	1796 (2.5)	2179 (2.6)	1173 (2.7)
Median Income (Residential area), $			
<38,000	11,070 (15.6)	13,997 (16.5)	7735 (17.6)
38,000–47,999	15,981 (22.5)	19,050 (22.5)	10,314 (23.4)
48,000–62,999	18,484 (26.0)	22,577 (26.6)	11,621 (26.4)
63,000+	25,008 (35.2)	28,566 (33.7)	14,044 (31.9)
Unknown	603 (0.8)	601 (0.7)	325 (0.7)
Without high school degree (Residential area), %			
<7	19,002 (26.7)	22,131 (26.1)	10,786 (24.5)
7–12.9	24,163 (34.0)	28,382 (33.5)	14,705 (33.4)
13–20.9	17,266 (24.3)	21,002 (24.8)	11,317 (25.7)
21+	10,155 (14.3)	12,677 (15.0)	6947 (15.8)
Unknown	560 (0.8)	549 (0.6)	284 (0.6)
Distance from facility to residence, miles			
<5	19,491 (27.4)	24,129 (28.5)	13,336 (30.3)
5–9.9	16,057 (22.6)	19,204 (22.7)	10,066 (22.9)
10–24.9	19,209 (27.0)	23,016 (27.2)	11,998 (27.2)
25+	15,849 (22.3)	17,877 (21.1)	8369 (19.0)
Unknown	540 (0.8)	515 (0.6)	270 (0.6)
Facility Type			
Non-academic	50,581 (71.1)	59,266 (69.9)	31,141 (70.7)
Academic/research	20,552 (28.9)	25,472 (30.1)	12,895 (29.3)
Unknown	13 (0.0)	3 (0.0)	3 (0.0)
Facility Location			
Northeast	17,655 (24.8)	20,439 (24.1)	10,348 (23.5)
South	24,678 (34.7)	27,877 (32.9)	15,175 (34.5)
Midwest	18,267 (25.7)	22,876 (27.0)	12,336 (28.0)
West	10,533 (14.8)	13,546 (16.0)	6177 (14.0)
Unknown	13 (0.0)	3 (0.0)	3 (0.0)
T-Stage			
1	22,681 (31.9)	3547 (4.2)	1420 (3.2)
2	45,633 (64.1)	40,791 (48.1)	16,698 (37.9)
2A	1886 (2.7)	36,115 (42.6)	2228 (5.1)
2B	943 (1.3)	4278 (5.0)	18,042 (41.0)
3	2 (0.0)	10 (0.0)	5385 (12.2)
4	1 (0.0)	-	266 (0.6)
PSA (Prostate-specific antigen), ng/mL			
0.2–2.9	7101 (10.0)	5063 (6.0)	1797 (4.1)
3.0–6.9	48,464 (68.1)	39,696 (46.8)	10,271 (23.3)
7.0–10.0	15,581 (21.9)	17,498 (20.6)	6488 (14.7)
10.1–12.9	-	12,501 (14.8)	3871 (8.8)
13.0–16.9	-	7131 (8.4)	2963 (6.7)
17.0–20.0	-	2852 (3.4)	1484 (3.4)
20.1–49.9	-	-	11,625 (26.4)
50.0–74.9	-	-	3850 (8.7)
>74.9	-	-	1690 (3.8)
Total Gleason Score			
<6	1732 (2.4)	8 (0.20)	1 (0.20)
6	69,414 (97.6)	401 (0.5)	145 (0.3)
7	-	13,523 (16.0)	4519 (10.3)
8	-	70,817 (83.6)	8325 (18.9)
9	-	-	18,490 (42.0)
10	-	-	11,450 (26.0)
Radiotherapy			
IMRT and No RT and IMRT Boost	25,235 (35.5)	48,160 (56.8)	32,851 (74.6)
IMRT and BT Boost	1604 (2.3)	7092 (8.4)	4038 (9.2)
Proton	1777 (2.5)	2296 (2.7)	488 (1.1)
SBRT	2918 (4.1)	3795 (4.5)	820 (1.9)
BT LDR	26,311 (37.0)	15,534 (18.3)	3607 (8.2)
BT HDR	13,301 (18.7)	7864 (9.3)	2235 (5.1)
Hormonal Therapy			
No	58,333 (82.8)	52,875 (62.4)	10,457 (23.7)
Yes	10,983 (15.4)	30,160 (35.6)	33,031 (75.0)
Unknown	1830 (2.6)	1706 (2.0)	1509 (2.4)

**Table 3 cancers-14-05503-t003:** Comparative utilization of the six treatment modalities for low-risk prostate cancer patients.

Low Risk	IMRT Only (*n* = 25,235)	IMRT + BT Boost (*n* = 1604)	Proton (*n* = 1777)	SBRT (*n* = 2918)	BT LDR (*n* = 26,311)	BT HDR (*n* = 13,301)	*p*-Value
Year of Diagnosis							<0.0001
2004–2007	9510 (37.7)	821 (51.2)	732 (41.2)	428 (14.7)	12,561 (47.7)	6935 (52.1)	
2008–2011	10,277 (40.7)	563 (35.1)	712 (40.1)	1148 (39.3)	9147 (34.8)	4580 (34.4)	
2012–2015	5448 (21.6)	220 (13.7)	333 (18.7)	1342 (46.0)	4603 (17.5)	1786 (13.4)	
Age (years)							<0.0001
<65	8629 (34.2)	794 (49.5)	892 (50.2)	1205 (41.3)	11,976 (45.5)	6305 (47.4)	
65–69	6438 (25.5)	391 (24.4)	490 (27.6)	789 (27.0)	6663 (25.3)	3188 (24.0)	
70–74	6008 (23.8)	278 (17.3)	272 (15.3)	596 (20.4)	5080 (19.3)	2526 (19.0)	
>74	4160 (16.5)	141 (8.8)	123 (6.9)	328 (11.2)	2592 (9.9)	1282 (9.6)	
Race							<0.0001
White	20,448 (81.0)	1211 (75.5)	1644 (92.5)	2460 (84.3)	22,437 (85.3)	10,954 (82.4)	
Black	3919 (15.5)	340 (21.2)	79 (4.4)	365 (12.5)	3195 (12.1)	1767 (13.3)	
Other	592 (2.3)	39 (2.4)	49 (2.8)	60 (2.1)	416 (1.6)	377 (2.8)	
Unknown (794)	276 (1.1)	14 (0.9)	5 (0.3)	33 (1.1)	263 (1.0)	203 (1.5)	
Insurance Status							<0.0001
None	317 (1.3)	19 (1.20)	44 (2.5)	25 (0.9)	189 (0.7)	106 (0.8)	
Government	15,594 (61.8)	810 (50.5)	804 (45.2)	1626 (55.7)	13,538 (51.5)	6514 (49.0)	
Private	8934 (35.4)	757 (47.2)	926 (52.1)	1215 (41.6)	12,262 (46.6)	6520 (49.0)	
Unknown (946)	390 (1.5)	18 (1.1)	3 (0.2)	52 (1.8)	322 (1.2)	161 (1.2)	
Charlson–Deyo Comorbidity index							<0.0001
0	22,238 (88.1)	1420 (88.5)	1613 (90.8)	2584 (88.6)	22,780 (86.6)	11,730 (88.2)	
1	2518 (10.0)	161 (10.0)	147 (8.3)	304 (10.4)	3086 (11.7)	1409 (10.6)	
2+	479 (1.9)	23 (1.4)	17 (1.0)	30 (1.0)	445 (1.7)	162 (1.2)	
Residential Setting							<0.0001
Metro	20,792 (84.3)	1360 (87.0)	1501 (87.9)	2542 (89.8)	20,267 (79.3)	11,145 (85.5)	
Urban	3460 (14.0)	189 (12.1)	190 (11.1)	259 (9.1)	4622 (18.1)	1673 (12.8)	
Rural	400 (1.6)	15 (1.0)	17 (1.0)	30 (1.1)	674 (2.6)	214 (1.6)	
Missing (1796)	-	-	-	-	-	-	
Median Income (Residential area), $							<0.0001
<38,000	4326 (17.3)	326 (20.6)	121 (6.9)	286 (9.9)	4211 (16.2)	1800 (13.6)	
38,000–47,999	5780 (23.1)	302 (19.1)	303 (17.2)	412 (14.2)	6425 (24.7)	2759 (20.9)	
48,000–62,999	6828 (27.2)	331 (21.0)	481 (27.3)	628 (21.7)	6923 (26.6)	3293 (24.9)	
63,000+	8129 (32.4)	620 (39.3)	858 (48.7)	1568 (54.2)	6478 (32.6)	5355 (40.5)	
Missing (603)	-	-	-	-	-	-	
Without high school degree (Residential area), %							<0.0001
<7	6248 (24.9)	386 (24.4)	696 (39.5)	1045 (36.1)	6686 (25.7)	3941 (29.8)	
7–12.9	8642 (34.5)	493 (31.2)	550 (31.2)	911 (31.5)	9117 (35.0)	4450 (33.7)	
13–20.9	6293 (25.1)	397 (25.1)	299 (17.0)	599 (20.7)	6498 (24.9)	3180 (24.1)	
21+	3892 (15.5)	303 (19.2)	218 (12.4)	341 (11.8)	3757 (14.4)	1644 (12.4)	
Missing (560)	-	-	-	-	-	-	
Distance from facility to residence, miles							<0.0001
<5	8633 (34.4)	561 (35.5)	72 (4.1)	573 (19.8)	6366 (24.4)	3286 (27.6)	
5–9.9	6408 (25.5)	401 (25.4)	71 (4.0)	656 (22.7)	5451 (20.9)	3070 (22.7)	
10–24.9	6828 (27.2)	408 (25.8)	151 (8.6)	852 (29.5)	7331 (28.1)	3639 (27.2)	
25+	3212 (12.8)	209 (13.2)	1472 (83.4)	812 (28.1)	6918 (26.5)	3226 (22.4)	
Missing (540)	-	-	-	-	-	-	
Facility Type							<0.0001
Non-academic	18,695 (74.1)	1339 (83.5)	48 (2.70)	1448 (49.6)	19,941 (75.8)	9110 (68.5)	
Academic/research	6538 (25.9)	264 (16.5)	1728 (97.3)	1470 (50.4)	6365 (24.2)	4187 (31.5)	
Missing (13)							
Facility Location							<0.0001
Northeast	7178 (28.4)	567 (35.4)	109 (6.1)	1190 (40.8)	5696 (21.7)	2915 (21.9)	
South	7954 (31.5)	721 (45.0)	36 (2.0)	1042 (35.7)	10,091 (38.4)	4834 (36.4)	
Midwest	7315 (29.0)	221 (13.8)	23 (1.3)	506 (17.3)	7022 (26.7)	3180 (23.9)	
West	2786 (11.0)	94 (5.9)	1608 (90.5)	180 (6.2)	3497 (13.3)	2368 (17.8)	
Missing (13)	-	-	-	-	-	-	
T-Stage *							<0.0001
1	8797 (34.9)	403 (25.1)	633 (35.6)	1775 (60.8)	7923 (30.1)	3150 (23.7)	
2	15,243 (60.4)	1129 (70.4)	1098 (61.8)	939 (32.2)	17,541 (66.7)	9683 (72.8)	
2A	834 (3.3)	58 (3.6)	40 (2.3)	148 (5.1)	502 (1.9)	304 (2.3)	
2B	359 (1.4)	14 (0.9)	6 (0.3)	56 (1.9)	345 (1.3)	163 (1.2)	
PSA (Prostate-specific antigen), ng/mL							<0.0001
0.2–2.9	2313 (9.20)	181 (11.3)	167 (9.4)	277 (9.5)	2714 (10.3)	1449 (10.9)	
3.0–6.9	16,538 (65.5)	1121 (69.9)	1225 (68.9)	1984 (68.0)	18,333 (69.7)	9263 (69.6)	
7.0–10.0	6384 (25.3)	302 (18.8)	385 (21.7)	657 (22.5)	5264 (20.0)	2589 (19.5)	
Total Gleason Score							<0.0001
<6	564 (2.2)	25 (1.6)	39 (2.2)	22 (0.8)	722 (2.7)	360 (2.7)	
6	24,671 (97.8	1579 (98.4)	1738 (97.8)	2896 (99.2)	25,589 (97.3)	12,941 (97.3)	
Hormonal Therapy							<0.0001
No	21,091 (85.9)	1079 (69.3)	1671 (96.0)	2630 (93.6)	20,999 (81.8)	10,863 (83.7)	
Yes	3468 (14.1)	477 (30.7)	69 (4.0)	180 (6.4)	4666 (18.2)	2123 (16.3)	
Unknown (1830)	-	-	-	-	-	-	

* *n* = 2 for stage 3 and *n* = 1 for stage 4.

**Table 4 cancers-14-05503-t004:** Comparative utilization of the six treatment modalities for intermediate-risk prostate cancer patients.

Intermediate Risk	IMRT Only (*n* = 48,160)	IMRT + BT Boost (*n* = 7092)	Proton (*n* = 2296)	SBRT (*n* = 3795)	BT LDR (*n* = 15,534)	BT HDR (*n* = 7864)	*p*-Value
Year of Diagnosis							<0.0001
2004–2007	12,498 (26.0)	2144 (30.2)	721 (31.4)	211 (5.6)	5579 (35.9)	3212 (40.8)	
2008–2011	18,438 (38.3)	2733 (38.5)	814 (35.5)	1240 (32.7)	5167 (33.3)	2627 (33.4)	
2012–2015	17,224 (35.8)	2215 (31.2)	761 (33.1)	2344 (61.8)	4788 (30.8)	2025 (25.8)	
Age (years)							<0.0001
<65	11,689 (24.3)	2734 (38.6)	852 (37.1)	1154 (30.4)	5303 (34.1)	2964 (37.7)	
65–69	10,952 (22.7)	1814 (25.6)	649 28.3)	973 (25.6)	3814 (24.6)	1958 (24.9)	
70–74	12,552 (26.1)	1566 (22.1)	464 (20.2)	924 (24.3)	3590 (23.1)	1680 (21.4)	
>74	12,967 (26.9)	978 (13.8)	331 (14.4)	744 (19.6)	2827 (18.2)	1262 (16.0)	
Race							<0.0001
White	38,718 (80.4)	5482 (77.3)	2079 (90.5)	3092 (81.5)	13,002 (83.7)	6351 (80.8)	
Black	7684 (16.0)	1322 (18.8)	125 (5.4)	571 (15.0)	2077 (13.4)	1131 (14.4)	
Other	1256 (2.6)	217 (3.1)	80 (3.50)	91 (2.4)	307 (2.0)	266 (3.4)	
Unknown (880)	502 (1.00)	61 (0.9)	12 (0.50)	41 (1.1)	148 (1.0)	116 (1.5)	
Insurance Status							<0.0001
None	615 (1.3)	58 (0.8)	62 (2.7)	31 (0.8)	117 (0.8)	70 (0.9)	
Government	33,895 (70.4)	4087 (57.6)	1332 (58.0)	2359 (62.2)	9415 (60.6)	4478 (56.9)	
Private	13,018 (27.0)	2869 (40.5)	887 (38.6)	1301 (34.3)	5838 (37.6)	3227 (41.0)	
Unknown (1082)	632 (1.3)	78 (1.1)	15 (0.7)	104 (2.7)	164 (1.1)	89 (1.10)	
Charlson–Deyo Comorbidity index							<0.0001
0	41,173 (85.5)	6097 (86.0)	2015 (87.8)	3223 (84.9)	13,079 (84.2)	6809 (86.6)	
1	5582 (11.6)	865 (12.2)	249 (10.8)	489 (12.9)	2069 (13.3)	903 (11.5)	
2+	1405 (2.9)	130 (1.8)	32 (1.4)	83 (2.2)	386 (2.5)	152 (1.9)	
Residential Setting							<0.0001
Metro	38,924 (82.9)	5954 (85.7)	1928 (87.3)	3275 (88.8)	11,589 (76.8)	6424 (83.8)	
Urban	7112 (15.1)	878 (12.6)	257 (11.6)	369 (10.0)	2931 (19.4)	1070 (14.0)	
Rural	923 (2.0)	113 (1.6)	24 (1.1)	44 (1.2)	574 (3.8)	173 (2.3)	
Missing (2179)	-	-	-	-	-	-	
Median Income (Residential area), $							<0.0001
<38,000	8388 (17.5)	1227 (17.4)	210 (9.2)	412 (10.9)	2676 (17.4)	1084 (13.9)	
38,000–47,999	11,077 (23.1)	1586 (22.5)	454 (19.9)	524 (13.9)	3796 (24.7)	1613 (20.7)	
48,000–62,999	13,208 (27.6)	1724 (24.5)	655 (28.7)	791 (21.0)	4148 (27.0)	2001 (25.6)	
63,000+	15,198 (31.7)	2497 (35.5)	965 (42.3)	2046 (54.2)	4755 (30.9)	3105 (39.8)	
Missing (601)	-	-	-	-	-	-	
Without high school degree (Residential area), %							<0.0001
<7	11,946 (24.9)	1841 (26.2)	771 (33.8)	1354 (35.9)	3891 (25.3)	2328 (29.8)	
7–12.9	16,371 (34.2)	2259 (32.1)	760 (33.3)	1164 (30.8)	5198 (33.8)	2630 (33.7)	
13–20.9	12,131 (25.3)	1748 (24.8)	443 (19.4)	782 (20.7)	3971 (25.8)	1927 (24.7)	
21+	7451 (15.6)	1188 (16.9)	310 (13.6)	475 (12.6)	2333 (15.2)	920 (11.8)	
Missing (549)	-	-	-	-	-	-	
Distance from facility to residence, miles							<0.0001
<5	15,957 (33.3)	2117 (30.1)	107 (4.7)	780 (20.6)	3372 (21.9)	1796 (23.0)	
5–9.9	11,833 (24.7)	1674 (23.8)	91 (4.0)	820 (21.7)	3077 (20.0)	1709 (21.9)	
10–24.9	13,229 (27.6)	1996 (28.3)	243 (10.6)	1107 (29.3)	4315 (28.0)	2126 (27.2)	
25+	6894 (14.4)	1254 (17.8)	1844 (80.7)	1071 (28.3)	4633 (30.1)	2181 (27.9)	
Missing (515)	-	-	-	-	-	-	
Facility Type							<0.0001
Non-academic	35,425 (73.6)	5129 (72.3)	105 (4.60)	1600 (42.2)	11,667 (75.1)	5340 (67.9)	
Academic/research	12,734 (26.4)	1962 (27.7)	2190 (95.4)	2195 (57.8)	3867 (24.9)	2524 (32.1)	
Missing (3)	-	-	-	-	-	-	
Facility Location							<0.0001
Northeast	12,791 (26.6)	1725 (24.3)	134 (5.8)	1589 (41.9)	2749 (17.7)	1451 (18.5)	
South	15,046 (31.2)	2903 (40.9)	83 (3.6)	1186 (31.3)	5964 (38.4)	2695 (34.3)	
Midwest	13,849 (28.8)	1458 (20.6)	47 (2.0)	787 (20.7)	4633 (29.8)	2102 (26.7)	
West	6473 (13.4)	1005 (14.2)	2031 (88.5)	233 (6.1)	2188 (14.1)	1616 (20.5)	
Missing (3)	-	-	-	-	-	-	
T-Stage *							<0.0001
1	1802 (3.7)	216 (3.0)	255 (11.1)	244 (6.4)	735 (4.7)	295 (3.8)	
2	21,995 (45.7)	3684 (51.9)	1247 (54.3)	739 (19.5)	8320 (53.6)	4806 (61.1)	
2A	21,609 (44.9)	2807 (39.6)	687 (29.9)	2565 (67.6)	5955 (38.3)	2492 (31.7)	
2B	2747 (5.7)	385 (5.4)	107 (4.70)	247 (6.50)	522 (3.4)	270 (3.4)	
PSA (Prostate-specific antigen), ng/mL							<0.0001
0.2–2.9	2766 (5.7)	402 (5.7)	132 (5.7)	194 (5.1)	996 (6.4)	573 (7.3)	
3.0–6.9	20,908 (43.4)	3624 (51.1)	1191 (51.9)	1920 (50.6)	7976 (51.3)	4077 (51.8)	
7.0–10.0	10,451 (21.7)	1430 (20.2)	463 (20.2)	799 (21.1)	2918 (18.8)	1437 (18.3)	
10.1–12.9	7341 (15.2)	857 (12.1)	300 (13.1)	516 (13.6)	2377 (15.3)	1110 (14.1)	
13.0–16.9	4721 (9.8)	550 (7.8)	142 (6.2)	286 (7.5)	964 (6.2)	468 (6.0)	
17.0–20.0	1973 (4.1)	229 (3.2)	68 (3.0)	80 (2.1)	303 (2.0)	199 (2.5)	
Total Gleason Score							<0.0001
<6	188 (0.4)	16 (0.2)	3 (0.1)	4 (0.1)	126 (0.8)	64 (0.8)	
6	6946 (14.4)	735 (10.4)	501 (21.8)	493 (13.0)	3205 (20.6)	1643 (20.9)	
7	41,026 (85.2)	6341 (89.4)	1792 (78.0)	3298 (86.9)	12,203 (78.6)	6157 (78.3)	
Hormonal Therapy							<0.0001
No	26,276 (55.6)	4242 (61.1)	1965 (87.3)	3137 (85.9)	11,308 (74.3)	5947 (77.3)	
Yes	21,016 (44.4)	2698 (38.9)	285 (12.7)	515 (14.1)	3904 (25.7)	1742 (22.7)	
Unknown (1706)	-	-	-	-	-	-	

* *n* = 10 for stage 3 and *n* = 0 for stage 4.

**Table 5 cancers-14-05503-t005:** Comparative utilization of the six treatment modalities for high-risk prostate cancer patients.

High Risk	IMRT Only (*n* = 32,851)	IMRT + BT Boost (*n* = 4038)	Proton (*n* = 488)	SBRT (*n* = 820)	BT LDR (*n* = 3607)	BT HDR (*n* = 2235)	*p*-Value
Year of Diagnosis							<0.0001
2004–2007	7246 (22.1)	1015 (25.1)	150 (30.7)	36 (4.4)	1507 (41.8)	867 (38.8)	
2008–2011	12,337 (37.6)	1564 (38.7)	112 (23.0)	311 (37.9)	1204 (33.4)	797 (35.7)	
2012–2015	13,268 (40.4)	1459 (36.1)	226 (46.3)	473 (57.7)	896 (24.8)	571 (25.5)	
Age (years)							<0.0001
<65	7094 (21.6)	1295 (32.1)	119 (24.4)	203 (24.8)	1154 (32.0)	738 (33.0)	
65–69	6541 (19.9)	961 (23.8)	123 (25.2)	176 (21.5)	874 (24.2)	541 (24.2)	
70–74	7965 (24.2)	1012 (25.1)	113 (23.2)	182 (22.2)	816 (22.6)	518 (23.2)	
>74	11,251 (34.2)	770 (19.1)	133 (27.3)	259 (31.6)	763 (21.2)	438 (19.6)	
Race							<0.0001
White	26,207 (79.8)	3069 (76.0)	417 (85.5)	667 (81.3)	2982 (82.7)	1789 (80.0)	
Black	5401 (16.4)	757 (18.7)	52 (10.7)	123 (15.0)	513 (14.2)	355 (15.9)	
Other	946 (2.9)	175 (4.3)	17 (3.5)	14 (1.7)	71 (2.0)	69 (3.1)	
Unknown (415)	297 (0.9)	37 (0.9)	2 (0.4)	16 (2.0)	41 (1.1)	22 (1.0)	
Insurance Status							<0.0001
None	518 (1.6)	36 (0.9)	7 (1.4)	9 (1.1)	24 (0.7)	23 (1.0)	
Government	24,018 (73.1)	2510 (62.2)	326 (66.8)	566 (69.0)	2316 (64.2)	1323 (59.2)	
Private	7912 (24.1)	1452 (36.0)	152 (31.1)	229 (27.9)	1224 (33.9)	854 (38.2)	
Unknown (540)	403 (1.2)	40 (1.0)	3 (0.6)	16 (2.0)	43 (1.2)	35 (1.6)	
Charlson–Deyo Comorbidity index							<0.0001
0	27,820 (84.7)	3428 (84.9)	426 (87.3)	689 (84.0)	3081 (85.4)	1902 (85.1)	
1	4001 (12.2)	517 (12.8)	53 (10.9)	113 (13.8)	442 (12.3)	287 (12.8)	
2+	1030 (3.1)	93 (2.3)	9 (1.8)	18 (2.2)	84 (2.3)	46 (2.1)	
Residential Setting							<0.0001
Metro	26,116 (81.7)	3418 (86.2)	418 (88.4)	700 (88.5)	2625 (74.7)	1774 (82.2)	
Urban	5134 (16.1)	481 (12.1)	50 (10.6)	87 (11.0)	753 (21.4)	328 (15.2)	
Rural	713 (2.2)	65 (1.6)	5 (1.1)	4 (0.5)	138 (3.9)	57 (2.6)	
Missing (1173)	-	-	-	-	-	-	
Median Income (Residential area), $							<0.0001
<38,000	5926 (18.2)	612 (15.3)	51 (10.5)	78 (9.70)	710 (19.8)	358 (16.2)	
38,000–47,999	7897 (24.2)	784 (19.6)	94 (19.4)	120 (14.9)	922 (25.8)	497 (22.4)	
48,000–62,999	8793 (27.0)	1061 (26.5)	151 (31.1)	173 (21.4)	876 (24.5)	567 (25.6)	
63,000+	10,009 (30.7)	1548 (38.7)	189 (39.0)	436 (54.0)	1069 (29.9)	793 (35.8)	
Missing (325)	-	-	-	-	-	-	
Without high school degree (Residential area), %							<0.0001
<7	7910 (24.2)	1080 (26.9)	135 (27.8)	284 (35.0)	790 (22.1)	587 (26.5)	
7–12.9	10,970 (33.6)	1343 (33.5)	147 (30.3)	225 (27.7)	1248 (34.9)	772 (34.8)	
13–20.9	8558 (26.2)	963 (24.0)	126 (26.0)	198 (24.4)	924 (25.8)	548 (24.7)	
21+	5218 (16.0)	623 (15.5)	77 (15.9)	104 (12.8)	616 (17.2)	309 (13.9)	
Missing (284)	-	-	-	-	-	-	
Distance from facility to residence, miles							<0.0001
<5	10,635 (32.6)	1238 (30.9)	38 (7.8)	154 (18.9)	768 (21.5)	503 (22.7)	
5–9.9	7778 (23.8)	903 (22.5)	46 (9.5)	165 (20.3)	707 (19.8)	467 (21.1)	
10–24.9	8887 (27.2)	1133 (28.3)	81 (16.7)	249 (30.6)	1016 (28.4)	632 (28.5)	
25+	5364 (16.4)	736 (18.4)	320 (66.0)	245 (30.1)	1088 (30.4)	616 (27.8)	
Missing (270)	-	-	-	-	-	-	
Facility Type							<0.0001
Non-academic	23,780 (72.4)	2811 (69.6)	43 (8.8)	325 (39.6)	2666 (73.9)	1516 (67.8)	
Academic/research	9069 (27.6)	1227 (30.4)	444 (91.2)	495 (60.4)	941 (26.1)	719 (32.2)	
Missing (3)	-	-	-	-	-	-	
Facility Location							<0.0001
Northeast	8174 (24.9)	814 (20.2)	58 (11.9)	372 (45.4)	542 (15.0)	388 (17.4)	
South	10,742 (32.7)	1571 (38.9)	35 (7.2)	269 (32.8)	1697 (47.0)	861 (38.5)	
Midwest	9561 (29.1)	910 (22.5)	22 (4.5)	134 (16.3)	1047 (29.0)	662 (29.6)	
West	4372 (13.3)	743 (18.4)	372 (76.4)	45 (5.5)	321 (8.9)	324 (14.5)	
Missing (3)	-	-	-	-	-	-	
T-Stage							<0.0001
1	940 (2.90)	79 (2.0)	20 (4.1)	83 (10.1)	173 (4.8)	125 (5.6)	
2	11,498 (35.0)	1568 (38.8)	204 (41.8)	140 (17.1)	2058 (57.1)	1230 (55.0)	
2A	1649 (5.0)	156 (3.9)	24 (4.9)	99 (12.1)	179 (5.0)	121 (5.4)	
2B	14,215 (43.3)	1647 (40.8)	161 (33.0)	469 (57.2)	974 (27.0)	576 (25.8)	
3	4307 (13.1)	577 (14.3)	78 (16.0)	29 (3.5)	215 (6.0)	179 (8.0)	
4	242 (0.7)	11 (0.3)	1 (0.2)	0	8 (0.2)	4 (0.2)	
PSA (Prostate-specific antigen), ng/mL							<0.0001
0.2–2.9	1396 (4.2)	160 (4.0)	21 (4.3)	23 (2.8)	121 (3.4)	76 (3.4)	
3.0–6.9	7590 (23.1)	1281 (31.7)	133 (27.3)	170 (20.7)	663 (18.4)	434 (19.4)	
7.0–10.0	5058 (15.4)	673 (16.7)	93 (19.1)	89 (10.9)	330 (9.1)	245 (11.0)	
10.1–12.9	3048 (9.3)	380 (9.4)	52 (10.7)	59 (7.2)	185 (5.1)	147 (6.6)	
13.0–16.9	2366 (7.2)	277 (6.9)	37 (7.6)	45 (5.5)	147 (4.1)	91 (4.1)	
17.0–20.0	1232 (3.8)	109 (2.7)	17 (3.5)	18 (2.2)	53 (1.5)	55 (2.5)	
20.1–49.9	8687 (26.4)	844 (20.9)	107 (21.9)	212 (25.9)	1115 (30.9)	660 (29.5)	
50.0–74.9	2399 (7.3)	255 (5.6)	15 (3.1)	150 (18.3)	669 (18.5)	392 (17.5)	
>74.9	1075 (3.3)	89 (2.2)	13 (2.7)	54 (6.6)	324 (9.0)	135 (6.0)	
Total Gleason Score							<0.0001
<6	86 (0.3)	8 (0.2)	1 (0.2)	2 (0.2)	37 (1.0)	11 (0.5)	
6	2320 (7.1)	217 (5.4)	39 (8.0)	174 (21.2)	1144 (31.7)	625 (28.0)	
7	6060 (18.4)	770 (19.1)	94 (19.3)	202 (24.6)	768 (21.3)	431 (19.3)	
8	14,208 (43.2)	1929 (47.8)	228 (46.7)	331 (40.4)	1080 (29.9)	714 (31.9)	
9	9257 (28.2)	1035 (25.6)	118 (24.2)	99 (12.1)	519 (14.4)	422 (18.9)	
10	920 (2.8)	79 (2.0)	8 (1.6)	12 (1.5)	59 (1.6)	32 (1.4)	
Hormonal Therapy							<0.0001
No	5755 (17.7)	955 (24.0)	187 (38.8)	530 (66.3)	1847 (52.2)	1183 (53.9)	
Yes	26,745 (82.3)	3019 (76.0)	295 (61.2)	270 (33.7)	1689 (47.8)	1013 (46.1)	
Unknown (1830)	-	-	-	-	-	-	

**Table 6 cancers-14-05503-t006:** Unadjusted and adjusted hazard ratios (HR) for respective univariate and multivariate Cox proportional hazard models of overall survival in low-risk prostate cancer patients who received one of the six treatments.

Factor	Univariable		Multivariable	
	HR (95% CI)	*p*-Value	HR (95% CI)	*p*-Value
Low-risk				
Year of Diagnosis				
2004–2007	1.0	-	1.0	-
2008–2011	1.01 (0.96–1.06)	0.839	1.04 (0.98–1.10)	0.235
2012–2015	0.96 (0.85–1.08)	0.480	1.06 (0.92–1.23)	0.413
Age (years)				
<65	1.0	-	1.0	-
65–69	1.73 (1.63–1.84)	<0.001	1.36 (1.27–1.47)	<0.001
70–74	2.42 (2.28–2.57)	<0.001	1.83 (1.70–1.97)	<0.001
>74	3.84 (3.61–4.07)	<0.001	2.86 (2.65–3.09)	<0.001
Race				
White	1.0	-	1.0	-
Black	1.03 (0.97–1.10)	0.322	1.05 (0.98–1.13)	0.157
Other	0.69 (0.58–0.82)	<0.001	0.79 (0.67–0.95)	0.010
Insurance Status				
None	1.0	-	1.0	-
Government	1.49 (1.18–1.90)	0.001	1.01 (0.78–1.31)	0.920
Private	0.65 (0.51–0.83)	0.001	0.72 (0.56–0.93)	0.011
Charlson–Deyo Comorbidity index				
0	1.0	-	1.0	-
1	1.61 (1.52–1.71)	<0.001	1.54 (1.44–1.64)	<0.001
2+	2.79 (2.47–3.15)	<0.001	2.59 (2.28–2.93)	<0.001
Residential Setting				
Metro	1.0	-	1.0	-
Urban	1.18 (1.11–1.25)	<0.001	1.06 (0.99–1.14)	0.086
Rural	1.38 (1.21–1.59)	<0.001	1.24 (1.07–1.43)	0.005
Median Income (Residential area), $				
<38,000	1.0	-	1.0	-
38,000–47,999	0.88 (0.82–0.94)	<0.001	0.92 (0.86–0.99)	0.027
48,000–62,999	0.78 (0.73–0.83)	<0.001	0.88 (0.82–0.95)	0.002
63,000+	0.63 (0.59–0.67)	<0.001	0.81 (0.74–0.88)	<0.001
Without high school degree (Residential area), %				
<7	1.0	-	1.0	-
7–12.9	1.23 (1.16–1.30)	<0.001	1.13 (1.06–1.20)	<0.001
13–20.9	1.41 (1.32–1.49)	<0.001	1.22 (1.13–1.32)	<0.001
21+	1.51 (1.41–1.62)	<0.001	1.25 (1.14–1.38)	<0.001
Distance from facility to residence, miles				
<5	1.0	-	1.0	-
5–9.9	0.94 (0.89–0.99)	0.027	1.02 (0.96–1.08)	0.590
10–24.9	0.90 (0.85–0.95)	<0.001	0.99 (0.93–1.05)	0.760
25+	0.83 (0.78–0.88)	<0.001	0.87 (0.81–0.94)	<0.001
Facility Type				
Non-academic	1.0	-	1.0	-
Academic/research	0.81 (0.77–0.85)	<0.001	0.95 (0.90–1.00)	0.060
Facility Location				
Northeast	1.0	-	1.0	-
South	1.14 (1.08–1.20)	<0.001	1.02 (0.96–1.08)	0.492
Midwest	1.08 (1.02–1.15)	0.006	0.97 (0.91–1.03)	0.271
West	0.67 (0.62–0.72)	<0.001	0.77 (0.71–0.84)	<0.001
T-Stage				
1	1.0	-	1.0	-
2	1.06 (0.98–1.13)	0.137	1.08 (0.99–1.19)	0.078
2A	1.13 (0.93–1.38)	0.206	1.13 (0.92–1.38)	0.239
2B	0.91 (0.71–1.17)	0.470	0.92 (0.71–1.19)	0.510
Radiotherapy				
IMRT only	1.0	-	1.0	-
IMRT + BT Boost	0.63 (0.54–0.73)	<0.001	0.71 (0.60–0.83)	<0.001
Proton	0.29 (0.23–0.37)	<0.001	0.51 (0.40–0.66)	<0.001
SBRT	0.74 (0.64–0.86)	<0.001	0.87 (0.75–1.02)	<0.001
BT LDR	0.75 (0.71–0.79)	<0.001	0.85 (0.81–0.90)	<0.001
BT HDR	0.69 (0.65–0.73)	<0.001	0.83 (0.78–0.89)	<0.001
Hormonal Therapy				
No	1.0	-	1.0	-
Yes	1.14 (1.08–1.19)	<0.001	0.97 (0.92–1.02)	0.256

**Table 7 cancers-14-05503-t007:** Unadjusted and adjusted hazard ratios (HR) for respective univariate and multivariate Cox proportional hazard models of overall survival in intermediate-risk prostate cancer patients who received one of the six treatments.

Factor	Univariable		Multivariable	
	HR (95% CI)	*p*-Value	HR (95% CI)	*p*-Value
Intermediate-risk				
Year of Diagnosis				
2004–2007	1.0	-	1.0	-
2008–2011	1.01 (0.97–1.05)	0.767	0.99 (0.95–1.04)	0.695
2012–2015	0.96 (0.89–1.03)	0.286	0.97 (0.88–1.06)	0.471
Age (years)				
<65	1.0	-	1.0	-
65–69	1.42 (1.34–1.51)	<0.001	1.18 (1.10–1.26)	<0.001
70–74	1.84 (1.74–1.95)	<0.001	1.47 (1.37–1.57)	<0.001
>74	2.85 (2.70–3.00)	<0.001	2.24 (2.10–2.39)	<0.001
Race				
White	1.0	-	1.0	-
Black	0.98 (0.93–1.03)	0.379	1.02 (0.96–1.08)	0.504
Other	0.72 (0.64–0.82)	<0.001	0.78 (0.68–0.90)	<0.001
Insurance Status				
None	1.0	-	1.0	-
Government	1.69 (1.37–2.08)	<0.001	1.29 (1.04–1.61)	0.023
Private	0.87 (0.71–1.08)	0.205	0.96 (0.77–1.19)	0.681
Charlson–Deyo Comorbidity index				
0	1.0	-	1.0	-
1	1.46 (1.39–1.54)	<0.001	1.47 (1.40–1.55)	<0.001
2+	2.32 (2.12–2.54)	<0.001	2.20 (2.00–2.42)	<0.001
Residential Setting				
Metro	1.0	-	1.0	-
Urban	1.15 (1.10–1.21)	<0.001	1.02 (0.96–1.08)	0.617
Rural	1.25 (1.12–1.40)	<0.001	1.24 (0.97–1.24)	0.130
Median Income (Residential area), $				
<38,000	1.0	-	1.0	-
38,000–47,999	0.93 (0.88–0.98)	0.004	0.96 (0.91–1.02)	0.220
48,000–62,999	0.84 (0.80–0.89)	<0.001	0.93 (0.87–0.99)	0.024
63,000+	0.67 (0.64–0.71)	<0.001	0.81 (0.75–0.88)	<0.001
Without high school degree (Residential area), %				
<7	1.0	-	1.0	-
7–12.9	1.19 (1.14–1.25)	<0.001	1.08 (1.02–1.14)	0.007
13–20.9	1.32 (1.25–1.39)	<0.001	1.15 (1.08–1.23)	<0.001
21+	1.41 (1.33–1.49)	<0.001	1.23 (1.14–1.33)	<0.001
Distance from facility to residence, miles				
<5	1.0	-	1.0	-
5–9.9	0.91 (0.86–0.95)	<0.001	0.96 (0.91–1.01)	0.104
10–24.9	0.88 (0.84–0.92)	<0.001	0.94 (0.90–0.99)	0.019
25+	0.83 (0.79–0.87)	<0.001	0.90 (0.84–0.95)	0.001
Facility Type				
Non-academic	1.0	-	1.0	-
Academic/research	0.82 (0.78–0.85)	<0.001	0.95 (0.90–0.99)	0.013
Facility Location				
Northeast	1.0	-	1.0	-
South	1.10 (1.05–1.16)	<0.001	1.04 (0.99–1.09)	0.134
Midwest	1.10 (1.05–1.16)	<0.001	1.02 (0.97–1.08)	0.392
West	0.76 (0.72–0.81)	<0.001	0.85 (0.80–0.91)	<0.001
T-Stage				
1	1.0	-	1.0	-
2	1.10 (0.98–1.24)	0.103	1.02 (0.90–1.16)	0.764
2A	1.09 (0.96–1.23)	0.186	1.00 (0.88–1.14)	0.948
2B	1.21 (1.04–1.41)	0.013	1.09 (0.93–1.27)	0.310
Radiotherapy				
IMRT only	1.0	-	1.0	-
IMRT + BT Boost	0.60 (0.56–0.65)	<0.001	0.70 (0.65–0.76)	<0.001
Proton	0.35 (0.30–0.41)	<0.001	0.56 (0.46–0.67)	<0.001
SBRT	0.76 (0.67–0.86)	<0.001	0.85 (0.75–0.97)	0.014
BT LDR	0.73 (0.70–0.77)	<0.001	0.81 (0.77–0.85)	<0.001
BT HDR	0.66 (0.62–0.70)	<0.001	0.80 (0.75–0.85)	<0.001
Hormonal Therapy				
No	1.0	-	1.0	-
Yes	1.14 (1.10–1.18)	<0.001	0.98 (0.94–1.02)	0.265

**Table 8 cancers-14-05503-t008:** Unadjusted and adjusted hazard ratios (HR) for respective univariate and multivariate Cox proportional hazard models of overall survival in high-risk prostate cancer patients who received one of the six treatments.

Factor	Univariable		Multivariable	
	HR (95% CI)	*p*-Value	HR (95% CI)	*p*-Value
High-risk				
Year of Diagnosis				
2004–2007	1.0	-	1.0	-
2008–2011	1.10 (1.05–1.16)	0.000	1.07 (1.02–1.14)	0.012 0.360
2012–2015	1.08 (0.99–1.17)	0.072	1.05 (0.95–1.16)	-
Age (years)				
<65	1.0	-	1.0	-
65–69	1.30 (1.21–1.40)	<0.001	1.16 (1.07–1.27)	<0.001
70–74	1.56 (1.45–1.67)	<0.001	1.35 (1.24–1.46)	<0.001
>74	2.41 (2.27–2.57)	<0.001	2.08 (1.92–2.25)	<0.001
Race				
White	1.0	-	1.0	-
Black	0.86 (0.81–0.91)	<0.001	0.92 (0.86–0.99)	0.023
Other	0.82 (0.71–0.94	0.005	0.89 (0.77–1.03)	0.107
Insurance Status				
None	1.0	-	1.0	-
Government	1.69 (1.37–2.08)	0.001	1.04 (0.84–1.29)	0.701
Private	0.87 (0.71–1.08)	0.157	0.87 (0.70–1.07)	0.184
Charlson–Deyo Comorbidity index				
0	1.0	-	1.0	-
1	1.31 (1.23–1.40)	<0.001	1.31 (1.23–1.40)	<0.001
2+	2.14 (1.92–2.40)	<0.001	2.09 (1.86–2.35)	<0.001
Residential Setting				
Metro	1.0	-	1.0	-
Urban	1.15 (1.09–1.22)	<0.001	1.08 (1.01–1.16)	0.031
Rural	1.11 (0.96–1.28)	0.161	1.06 (0.90–1.24)	0.482
Median Income (Residential area), $				
<38,000	1.0	-	1.0	-
38,000–47,999	1.03 (0.96–1.10)	0.426	1.01 (0.94–1.09)	0.797
48,000–62,999	0.94 (0.88–1.01)	0.075	0.95 (0.88–1.03)	0.230
63,000+	0.80 (0.75–0.85)	<0.001	0.87 (0.79–0.95)	0.003
Without high school degree (Residential area), %				
<7	1.0	-	1.0	-
7–12.9	1.13 (1.07–1.20)	<0.001	1.07 (1.01–1.15)	0.034
13–20.9	1.16 (1.09–1.23)	<0.001	1.09 (1.01–1.18)	0.030
21+	1.21 (1.13–1.30)	<0.001	1.17 (1.07–1.29)	0.001
Distance from facility to residence, miles				
<5	1.0	-	1.0	-
5–9.9	0.95 (0.89–1.00)	0.053	0.99 (0.93–1.05)	0.647
10–24.9	0.92 (0.87–0.97)	0.002	0.95 (0.90–1.01)	0.104
25+	0.86 (0.81–0.92)	<0.001	0.87 (0.81–0.94)	<0.001
Facility Type				
Non-academic	1.0	-	1.0	-
Academic/research	0.83 (0.79–0.87)	<0.001	0.94 (0.90–1.00)	0.033
Facility Location				
Northeast	1.0	-	1.0	-
South	1.09 (1.03–1.16)	0.003	1.10 (1.03–1.17)	0.003
Midwest	1.10 (1.04–1.17)	0.001	1.05 (0.99–1.12)	0.138
West	0.89 (0.83– 0.96)	0.002	0.90 (0.83–0.97)	0.008
T-Stage				
1	1.0	-	1.0	-
2	1.26 (1.07–1.49)	0.006	1.26 (1.05–1.51)	0.012
2A	1.46 (1.19–1.78)	<0.001	1.38 (1.12–1.71)	0.003
2B	1.33 (1.12–1.58)	0.001	1.25 (1.05–1.50)	0.015
3	1.48 (1.24–1.76)	<0.001	1.53 (1.27–1.84)	<0.001
4	2.46 (1.87–3.23)	<0.001	2.79 (2.10–3.70)	<0.001
Radiotherapy				
IMRT only	1.0	-	1.0	-
IMRT + BT Boost	0.59 (0.55–0.65)	<0.001	0.68 (0.62–0.74)	<0.001
Proton	0.48 (0.36–0.63)	<0.001	0.64 (0.48–0.87)	0.004
SBRT	0.73 (0.59–0.91)	0.005	0.86 (0.68–1.08)	0.187
BT LDR	0.66 (0.61–0.71)	<0.001	0.76 (0.69–0.82)	<0.001
BT HDR	0.62 (0.56–0.69)	<0.001	0.74 (0.66–0.82)	<0.001
Hormonal Therapy				
No	1.0	-	1.0	-
Yes	1.21 (1.15–1.27)	<0.001	1.05 (1.00–1.11)	0.062

## Data Availability

The data presented in this study are available in the National Cancer Database (NCDB).

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
