# Peer review of "Pattern of Radiotherapy Treatment in Low-Risk, Intermediate-Risk, and High-Risk Prostate Cancer Patients: Analysis of National Cancer Database"

_cancers, 2022, doi:10.3390/cancers14225503_

Round 1

Reviewer 1 Report

In this study the authors retrospectively compared the utilization rates and survival outcomes of different radiotherapy techniques for prostate cancer. They found that among six investigated modalities, IMRT had the lowest survival probability in all risk categories, while the highest survival probability was observed for the Proton therapy.

Radiotherapy is a mainstay in modern anticancer therapy, especially for prostate cancer. Although it is a computer-intensive and technology-heavy discipline, it cannot be reduced to a level of a one-bit yes/no item, or to just a therapy that can be delivered with different techniques, like it is often recorded in tumor registries. Radiotherapy is a matter of dose, volume, fractionation and overall treatment time interacting with the complexity of a human body. The analysis herein presented precludes meaningful inference with regard to the optimal use of radiotherapy for prostate cancer and does not provide any help in guiding clinicians in their daily practice. It sounds like a “phishing expedition” in population-based registries, which resulted in a and useless, if not misleading, information to the readership.

Author Response

Point 1: Radiotherapy is a mainstay in modern anticancer therapy, especially for prostate cancer. Although it is a computer-intensive and technology-heavy discipline, it cannot be reduced to a level of a one-bit yes/no item, or to just a therapy that can be delivered with different techniques, like it is often recorded in tumor registries. Radiotherapy is a matter of dose, volume, fractionation and overall treatment time interacting with the complexity of a human body. The analysis herein presented precludes meaningful inference with regard to the optimal use of radiotherapy for prostate cancer and does not provide any help in guiding clinicians in their daily practice. It sounds like a “phishing expedition” in population-based registries, which resulted in a and useless, if not misleading, information to the readership.

Response 1: While we hope the findings of our paper can better illustrate the past trends in radiotherapy for PCa, we do not endorse the preference of any single treatment over another. Instead, we hope that awareness of past trends can help physicians and patients make better-informed decisions when considering all factors in treatment selection. We have included text in the Introduction and Conclusion to relay this disclaimer to readers and clarification of our aim of providing simply a description picture of radiotherapy management, rather than a goal-oriented one.

Reviewer 2 Report

This is an interesting retrospective observational study, with some limitations.

1) In the methods consider to define all the relevant covariates included in the cox model. Have you considered PSA?

2) It is curious to evaluate that a considerable proportion of high risk patients have PSA values between 3.0 and 6.9 ng/ml  and some patients between 0.2 and 2.9 ng/ml. I expected higher values.

A recent study has evaluated PSA levels predictive for advanced PCa and I expected fewer patients in the indicated range.

Ref: Definition of Outcome-Based Prostate-Specific Antigen (PSA) Thresholds for Advanced Prostate Cancer Risk Prediction. Cancers 2021, 13, 3381. 

How can you explain it? 

3) In the limitations consider that: PSA levels here reported may have been obtained by different methods, and calibrations and this may be a font of bias in the risk prediction 

4)Acronyms in the abstract should be spelled out.

Author Response

Point 1: In the methods consider to define all the relevant covariates included in the cox model. Have you considered PSA?

Response 1: Relevant covariates included in the cox model are years of diagnosis, age, race, insurance status, Charleson-Deyo morbidity index, residential setting, median income, distance from facility to residence, facility type, facility location, hormonal therapy, radiotherapy (the key independent variable). PSA is not considered in cox model as PSA scores are already included in the definition of risk groups.

Point 2: It is curious to evaluate that a considerable proportion of high risk patients have PSA values between 3.0 and 6.9 ng/ml and some patients between 0.2 and 2.9 ng/ml. I expected higher values.

A recent study has evaluated PSA levels predictive for advanced PCa and I expected fewer patients in the indicated range.

Ref: Definition of Outcome-Based Prostate-Specific Antigen (PSA) Thresholds for Advanced Prostate Cancer Risk Prediction. Cancers 2021, 13, 3381. 

How can you explain it? 

Response 2: We have followed NCCN guidelines for stratifying the risk group as defined below. We agree, there is a sizeable number of patients in the high-risk category with relatively lower PSA scores. However, we have revisited the NCDB data set and checked our analysis. We did not find any mistakes. 

  1. Low-risk: PSA <10 & GS ≤ 6 & T-stage (T1, T1a, T1b, T1c, T2a); also take “T2” who has PSA <10 & GS ≤ 6
  2. Intermediate-risk: PSA=10-20 or GS= 7 or T-stage (T2b, T2c); also take “T2” who has PSA=10-20 or GS= 7
  3. High-risk: PSA > 20 or GS= 8, 9,10 or T-stage (T3, T3a, T3b, T4)

Point 3: In the limitations consider that: PSA levels here reported may have been obtained by different methods, and calibrations and this may be a font of bias in the risk prediction 

Response 3: Thank you for acknowledging this potential limitation of a retrospective study. While this is largely an unavoidable factor in many other retrospective studies, it is important to include it for the readers. This has now been mentioned in the limitation section.

Point 4: Acronyms in the abstract should be spelled out.

Response 4: Acronyms in the abstract have now been fully spelled out prior to being used regularly.

Reviewer 3 Report

General Comments:

The introduction is clear and concise. Some detail on abbreviations only. Some information is required in the methods section as some vital information is lacking. The discussion and conclusion are clear and easy to understand with the relevant information and some points to substantial the claims.

The Abstract

The abstract does summarise the manuscript. It is also valid to note that the abstract can be understood, and it is clear and concise.

The Introduction

Overall provided extensive information and was well researched.

Provide EBRT and BT LDR in detail

The Methods Section

Under data source, is this data ethically clear? Provide details

Under subject selection – give details and a reason for the time frame. Why not longer or before 2004?

Line 135-137 page 3 – only suggest surgery however, figure 1 has chemo – please clarify

Can some of the data/graphs come under supplementary information?

The Results Section

The results are clearly explained and easy to follow through the presentations. The multivariable analyses – present very similar results and description/information as the univariable analyses – maybe consider only presenting the differences that were found.

The Discussion Section

The discussion is concise and clear. It is easy to follow the authors’ direction in explaining the different points. However, a little more work is needed to substantiate many of the claims.

The third paragraph page 22 – are there any intervention examples of proton therapy and probably explain in more detail why the author speculates as such?

Page 23 paragraph 2 – Line 374 and 375 – please explain what is the difference in the radiotherapy modalities to give a better context.

Similar to the above explain Line 381 and 382 – why is a low-risk patient not recommended hormonal therapy and this is still done – explain and elaborate.

Author Response

Point 1: General Comments:

The introduction is clear and concise. Some detail on abbreviations only. Some information is required in the methods section as some vital information is lacking. The discussion and conclusion are clear and easy to understand with the relevant information and some points to substantial the claims.

Response 1: Thank you for your comments.

Point 2: The Abstract

The abstract does summarise the manuscript. It is also valid to note that the abstract can be understood, and it is clear and concise.

Response 2: Thank you for your comments.

Point 3: The Introduction

Overall provided extensive information and was well researched.

Provide EBRT and BT LDR in detail

Response 3: Thank you for your comments. In this manuscript, we wanted to compare different radiation techniques that were used to treat prostate cancer, as prostate cancer can be treated using different radiation technology. IMRT and SBRT are two of the techniques of EBRT and LDR and HDR BT are two different ways of performing brachytherapy (BT). These techniques are self-explanatory as these are commonly practiced in the treatment of prostate cancer. This has been expanded in the abstract and in the text appropriately.

Point 4: The Methods Section

Under data source, is this data ethically clear? Provide details

Under subject selection – give details and a reason for the time frame. Why not longer or before 2004?

Line 135-137 page 3 – only suggest surgery however, figure 1 has chemo – please clarify

Can some of the data/graphs come under supplementary information?

Response 4: The “Disclaimer” paragraph in the final part of the Methods section states that our research was conducted in accordance with the guidelines, data dictionary, and de-identified files provided by the NCDB. Similar to other registry-based studies, our study has no ethical implications and simply relies on previously collected data available in the database.

We appreciate the comment. The current NCDB data includes data only from 2004. While it could be great to include data starting before 2004; unfortunately, it is not feasible to include patients prior to the year 2004.  

This has been clarified in the paragraph to reflect the exclusion criteria as mentioned in Figure 1.

We could move some of the graphs depicting radiotherapy utilization (Figures 2-4) to the supplementary section, though we initially felt that the data from these Figures were critical to explaining Aim #1 of the paper.

Point 5: The Results Section

The results are clearly explained and easy to follow through the presentations. The multivariable analyses – present very similar results and description/information as the univariable analyses – maybe consider only presenting the differences that were found.

Response 5: Thank you for your comments. This was considered during the original drafting of the paper. However, we considered it may be more beneficial to be thorough with a full discussion of the multivariable analysis, as this deals directly with Aim #2 of the paper. While it may appear redundant to the univariable analysis, we believed re-emphasizing the associations that certain variables held with overall survival may represent their significance to the reader.

Point 6: The Discussion Section

The discussion is concise and clear. It is easy to follow the authors’ direction in explaining the different points. However, a little more work is needed to substantiate many of the claims.

The third paragraph page 22 – are there any intervention examples of proton therapy and probably explain in more detail why the author speculates as such?

Page 23 paragraph 2 – Line 374 and 375 – please explain what is the difference in the radiotherapy modalities to give a better context.

Similar to the above explain Line 381 and 382 – why is a low-risk patient not recommended hormonal therapy and this is still done – explain and elaborate.

Response 6: We believed the reader would be able to connect the prior referenced statement (higher cost of PBT treatment) to our speculation. A higher cost of treatment would logically be afforded by higher socioeconomic/educational backgrounds. However, we note it would need to be further studied to make such claims, as a literature search could not find such a connection thus far.

We have added additional statements to give better context to this comment. Our statements include clarification of data from our study, as well as referenced findings from similar registry-based studies.

We appreciate this comment. We agree with the reviewer that the use of hormonal therapy is not commonly recommended for low-risk prostate cancer. The database expands from 2004 to 2017 during which prostate cancer screening, diagnosis, and treatment have evolved significantly. The NCDB data captures what is the general practice in the community since it is a large hospital-based registry. While we noted about 15% of patients in the low-risk group received hormonal therapy, the majority (i.e., about 85%) of patients did not. It seems to align with the current recommendation of not using hormonal therapy in this group.

Reviewer 4 Report

General comment

The manuscript entitled “Pattern of Radiotherapy Treatment in Low-Risk, Intermediate-Risk, and High-Risk Prostate Cancer Patients: Analysis of National Cancer Database” aims to evaluate, via a retrospective trend analysis, the role of radiotherapy in the treatment of patients with PCa. The manuscript is, first of all, well written and reports a great amount of data. I appreciated, in particular the structured pattern of your paper. Nevertheless, few corrections are required, in my opinion, in order to improve the already fair quality of the manuscript. Suggested corrections are reported below.

INTRODUCTION

Albeit the introduction is well written and comprehensive, it is maybe too long and verbose. Probably I would move something written here in the initial part of the discussion.

I would highlight the aim of your study and the importance of your trend analysis.

MATERIALS AND METHODS

137-139: I would move these sentences in the results paragraph.

148-155: Similarly to before, also in this case the size of groups would be better placed in the results.

RESULTS

This is a huge amount of data and this is one of the strengths of the study. However, tables should be probably revised in order to improve readability and clarity.

DISCUSSION

The discussion seems, in some parts, a mere repetition of the results. I suggest you to compare your findings with other similar studies and, most importantly, discuss also other minor findings which could be interesting for the reader (for example the differences related to financial status as well as a rural or urban residence).

The manuscript lacks of future perspectives, in particular, I would suggest you to add something related to the usefulness of radiomics in RT treatment. Please see: DOI: 10.1177/17562872221109020 and DOI: 10.1007/s00066-020-01679-9

TABLES

In table 1, why no statistical comparisons were made among groups?

Similarly to before, in table 2 statistical comparisons would be a nice addition, albeit not being primarily related to the aim of the study.

Author Response

Point 1: 

INTRODUCTION

Albeit the introduction is well written and comprehensive, it is maybe too long and verbose. Probably I would move something written here in the initial part of the discussion.

I would highlight the aim of your study and the importance of your trend analysis.

Response 1: Our aims have been more clearly updated in the introduction. The topics in the introduction help give the readers a precedent of the pros/cons held by each modality and a simplified version of past data regarding survival outcomes.

Point 2: 

MATERIALS AND METHODS

137-139: I would move these sentences in the results paragraph.

148-155: Similarly to before, also in this case the size of groups would be better placed in the results.

Response 2: The aforementioned sentences have been moved to the Results section.

Point 3: 

RESULTS

This is a huge amount of data and this is one of the strengths of the study. However, tables should be probably revised in order to improve readability and clarity.

Response 3: We have tried various presentations of the data tables and seem to find this as the most readable version. Perhaps, the structure of Tables 4A-C could be adjusted to reflect the full width of the paper in the final publication.

Point 4: 

DISCUSSION

The discussion seems, in some parts, a mere repetition of the results. I suggest you to compare your findings with other similar studies and, most importantly, discuss also other minor findings which could be interesting for the reader (for example the differences related to financial status as well as a rural or urban residence).

The manuscript lacks of future perspectives, in particular, I would suggest you to add something related to the usefulness of radiomics in RT treatment. Please see: DOI: 10.1177/17562872221109020 and DOI: 10.1007/s00066-020-01679-9

Response 4: Thank you for the suggestions. We do believe in emphasizing the critical results from the previous section and we have now included additional studies to compare some notable results (distance to facility, area affecting RT modality availability, radiomics, etc.)

Thank you for sharing this advancing field in the study of prostate cancer. After reading through various literature, we do believe that this could be investigated in future studies and have now acknowledged it in the discussion.

Point 5: 

TABLES

In table 1, why no statistical comparisons were made among groups?

Similarly to before, in table 2 statistical comparisons would be a nice addition, albeit not being primarily related to the aim of the study.

Response 5: We appreciate the comment. Tables 1 and 2 are purposeful in presenting the demographics of the studied population in this paper. We believe that investigating the statistical analysis may distract readers from the real aims of this study (investigation of characteristics and overall survival). Further statistical comparisons for the data in Table 2 are already present in Tables 3A-C, with stratification via risk category and radiotherapy modality. The statistical analysis that correlates with the findings in Table 2 is presented later in the paper with univariate and multivariate analysis, with the study of the hazard ratios (Tables 4A-C). It is important to be mindful that all these tables are largely descriptive and should be mindful of the inherent selection bias that is present in the database.

Round 2

Reviewer 1 Report

I don't think that correlation of survival with treatment techniques is appropriate, without having considered RT dose, volume, and fractionation.

Unlike the authors claim, I disagree that the information herein provided can help physicians and patients make better-informed decisions when considering all factors in treatment selection. Instead, it can me highly misleading.

Author Response

Point 1: I don't think that correlation of survival with treatment techniques is appropriate, without having considered RT dose, volume, and fractionation.

Unlike the authors claim, I disagree that the information herein provided can help physicians and patients make better-informed decisions when considering all factors in treatment selection. Instead, it can me highly misleading.

Response 1: Thank you for the comments. The main purpose of the paper is to describe the patterns of care over time for patients treated with various radiation modalities. Survival data is available in the NCDB, and so it is reported, but such survival differences per modality largely reflect selection bias rather than efficacy of treatment.  We agree that the correlation of survival with treatment techniques and attribution of such survival benefits to the modality is inappropriate, and we have made an effort to point out the presence of bias and confounding factors in the manuscript. Details regarding dose, fractionation, volume, etc. would not be expected to lead to discernable differences in overall survival, but we do report ranges of doses as reported in the database for various treatment regimens. We have edited the paper to remove language that suggests that this paper should guide clinical decision-making due to the concerns that you correctly point out. We do think the paper is valuable in describing trends in radiotherapy technique utilization in the US and describing survival outcomes in a real-world population. We have also included calls for increased randomized clinical trials in our conclusion. We hope you find the edited manuscript acceptable.

Reviewer 2 Report

The reference 2 is not correct for this sentence in the Discussion, 

"....... inherently misrepresented risk stratification of PCa patients built into the NCDB, due to 415 possible differences in PSA acquisition and calibration methods. (2)"

since the problems related to miscalibration and difference between PSA methods are reported in this paper:

 Ferraro S et al Serum prostate specific antigen (PSA) testing for early detection of prostate cancer: Managing the gap between clinical and laboratory practice. Clin Chem 2021;67:602-609.

 Furthermore I would correct "possible differences in PSA acquisition" in "likely difference between PSA results due to poor harmonization of the methods"

Author Response

Point 1: 

The reference 2 is not correct for this sentence in the Discussion, 

"....... inherently misrepresented risk stratification of PCa patients built into the NCDB, due to 415 possible differences in PSA acquisition and calibration methods. (2)"

since the problems related to miscalibration and difference between PSA methods are reported in this paper:

 Ferraro S et al Serum prostate specific antigen (PSA) testing for early detection of prostate cancer: Managing the gap between clinical and laboratory practice. Clin Chem 2021;67:602-609.

Response 1: Please note that (2) is not a reference - it is the 2nd point of the discussion. We have edited the sentence and added this reference (ref #40).

Point 2: Furthermore I would correct "possible differences in PSA acquisition" in "likely difference between PSA results due to poor harmonization of the methods"

Response 2: We agree with the comments. Unfortunately, the NCDB reflects real world practice, which is replete with such variation, and the details regarding PSA measurement are not recorded. The number of institutions represented in this report, even for modalities less commonly used, is sufficiently large that the effects of such inconsistency would have minimal impact.

Reviewer 4 Report

The authors improved the manuscript in according to suggestions. No further comments from my side.

Author Response

Point 1: The authors improved the manuscript in according to suggestions. No further comments from my side.

Response 1: Thank you for your comments.